# PROPERTY-ORIENTED AND STRUCTURALLY MINIMAL FEEDBACK FOR EFFECTIVE LLM CODE REFINEMENT

## ABSTRACT

Large Language Models (LLMs) excel at code generation, yet ensuring the functional correctness of their outputs remains a persistent challenge. While recent studies have applied Test-Driven Development (TDD) to refine code, these methods are fundamentally undermined by poor feedback quality, stemming from the scarcity of high-quality test cases and noisy signals from auto-generated ones. In this work, we shift the focus from test quantity to feedback quality. We introduce the Property-Generated Solver (PGS), a novel paradigm designed to generate highly effective feedback by adhering to two principles: it must be **property-oriented**, to provide semantic guidance beyond simple I/O mismatches, and **structurally minimal**, to reduce cognitive load and isolate the error's root cause. PGS operates by checking high-level program properties (*e.g.*, a sorting function must produce a non-decreasing sequence) and then providing the simplest failing counterexample to the LLM. This property-driven, minimal feedback steers LLMs toward more correct and generalizable solutions. Across a diverse suite of programming benchmarks, PGS consistently demonstrates a superior corrective power, achieving a bug fix rate 1.4x-1.6x higher than the strongest debugging-based approaches and establishing a new state-of-the-art in automated code refinement. The source code and data are available in the supplementary.

## 1 INTRODUCTION

Recent advances in Large Language Models (LLMs) have revolutionized automated code generation, enabling tools like GitHub Copilot to assist developers in translating natural language requirements into functional code (OpenAI, 2023; Bai et al., 2023; Zhu et al., 2024). However, ensuring the functional correctness of the generated code remains a critical challenge, representing a primary bottleneck to the reliable deployment of these models in real-world scenarios (Liu et al., 2024a). To bridge this gap, many have turned to the Test-Driven Development (TDD) (Jiang et al., 2023; Zhong et al., 2024; Shinn et al., 2024) framework for iterative refinement. TDD framework leverages a cycle of test execution and code modification, where outcomes like pass/fail status or error messages, serve as feedback to guide the LLM. This feedback loop allows the model to progressively debug and enhance its initial code generation, steering it toward a correct solution.

However, the practical effectiveness of this feedback loop is constrained by two fundamental problems. The first is the scarcity of high-quality test cases. While a straightforward remedy might be to use LLMs to generate more tests (Chen et al., 2023a; Liu et al., 2024b), this approach often leads to a "cycle of self-deception," where the test generator shares the same logical biases as the code generator. Furthermore, generating the correct output for a given test input (*a.k.a* test oracle) can be as difficult for an LLM as solving the original problem itself (Barr et al., 2014; Jain et al., 2025). Second, and more critically, the quality of the feedback itself has been largely overlooked. The mass generation of flawed tests, as mentioned, directly degrades feedback quality by providing an uninformative and noisy signal to the LLM. Moreover, even a test case that correctly identifies a bug may be detrimental if its feedback is too complex. As shown in Figure 1 (b), a lengthy, convoluted failing input generates an intricate execution trace. This creates a high cognitive load that can overwhelm the LLM's reasoning process and misguide its refinement attempts, leading to repair failure.

This raises a crucial research question: *How can we construct high-quality, actionable feedback within the TDD framework to enable robust code refinement for LLMs?* Our answer is built on a

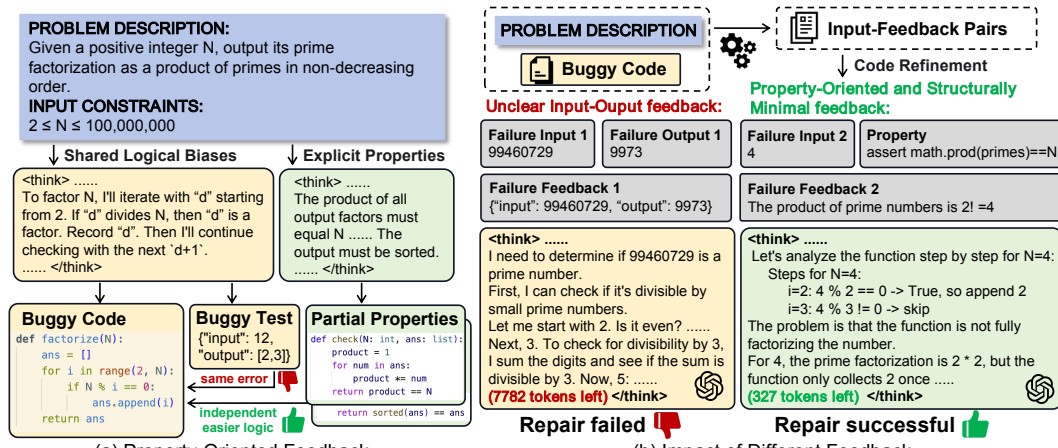

Figure 1: **The Principles of Effective Feedback for LLM Code Refinement. (a)** Property-driven validation avoids the pitfall of shared logical biases. While a buggy code generator often produces equally flawed I/O tests, property checks rely on simpler, independent logic, providing a more reliable signal for correctness. **(b)** A comparison of feedback impact. A complex, I/O-based counterexample creates high cognitive load and leads to repair failure. In contrast, our **property-oriented** and **structurally minimal** feedback uses the simplest failing input to provide a clear, actionable signal (*e.g.*, "The product of prime numbers is 2 != 4"), enabling a successful repair.

core insight: the most effective feedback is not found by generating more tests, but by improving the intrinsic quality of feedback itself. We propose that ideal feedback must be both **property-oriented** and **structurally minimal**, shifting the focus from finding bugs incidentally through mass test generation to a more deliberate process of semantic validation, as shown in Figure 1 (a). Specifically, we define these two principles as follows: **(1) Property-Oriented.** The feedback should move beyond concrete input-output pairs to validate fundamental program properties (Claessen & Hughes, 2000), *i.e.*, high-level, universal characteristics the code must satisfy for all valid inputs. For instance, a key property of any sorting function is that its output must be a non-decreasing sequence. This form of abstract validation not only circumvents the difficult test oracle problem but, more importantly, provides clear semantic guidance that helps the LLM generalize its repair. **(2) Structurally Minimal.** Rather than using a raw, complex failing test case, the feedback should be derived from the simplest possible counterexample that violates a property. This minimality isolates the error's root cause, removes distracting noise from the execution trace, and provides a clean, actionable signal tailored to an LLM's step-by-step reasoning. This directly addresses the cognitive load issue, preventing the model from being overwhelmed by convoluted failure scenarios (Liu et al., 2024c).

To operationalize these principles, we introduce the Property-Generated Solver (PGS), a multi-agent framework for feedback-centric code refinement. PGS employs a collaborative loop between a **Generator** agent, which produces and refines code, and a **Tester** agent, which crafts the property-oriented and structurally minimal feedback. Specifically, the Tester validates the generated code against high-level properties derived from the problem description. Upon detecting a violation, it identifies the structurally minimal counterexample and formulates it into a concise, actionable signal for the Generator. This loop of targeted feedback and iterative refinement steers the solution toward functional correctness and generalizability. Across a diverse suite of benchmarks, from function-level synthesis (Chen et al., 2021; Austin et al., 2021) to complex competition-level problems (Jain et al., 2025; Li et al., 2022) and repository-level tasks (Jimenez et al., 2024), PGS consistently establishes a new state-of-the-art, demonstrating a superior corrective power by achieving a bug fix rate **1.4x-1.6x higher** than even the most sophisticated debugging frameworks.

Our main contributions can be summarized as below:

- **A New Principle for Feedback Design.** We establish that effective feedback for LLM-based code refinement must be both property-oriented and structurally minimal. This shifts the focus from test quantity to feedback quality, addressing a critical dimension overlooked by existing TDD methods that often produce noisy and convoluted signals.

- **A Novel Multi-Agent Framework.** We design and implement the Property-Generated Solver (PGS), a multi-agent framework that operationalizes the proposed principles. PGS utilizes a collaborative loop between a Generator and a Tester agent to systematically construct high-quality feedback and guide the iterative refinement process.
- **State-of-the-Art Empirical Results.** We demonstrate through extensive experiments on multiple challenging benchmarks that PGS significantly outperforms existing TDD-based methods. Our framework sets a new state-of-the-art for automated code refinement, showcasing the benefits of high-quality feedback.

## 2 RELATED WORK

**LLM-Driven Code Refinement.** A significant research stream improves LLM-generated code through iterative refinement. Inspired by software development workflows (Jin et al., 2024; Xia & Zhang, 2023), these approaches guide LLMs using feedback from program execution. Techniques range from direct prompting with error messages, as in Self-Edit (Zhang et al., 2023) and Reflexion (Shinn et al., 2024), to complex debugging pipelines that integrate tools like static analyzers and debuggers (Zhong et al., 2024; Shi et al., 2024). While these methods have proven that external feedback is crucial for enhancing code generation, their effectiveness is fundamentally limited by the availability and quality of the test cases that produce this feedback. Our work addresses this core limitation by proposing a principled way to generate high-quality, targeted feedback when initial tests are sparse or non-existent.

**Specification-Based Test and Property Generation.** To overcome test scarcity, one line of research uses LLMs to generate I/O test cases (Chen et al., 2023a; Liu et al., 2024b), but this faces challenges like the "cycle of self-deception" and the oracle problem (Barr et al., 2014; Jain et al., 2025). A more promising direction is generating high-level properties. Prior work has shown LLMs can generate formal specifications (Endres et al., 2024) or properties for program verification (Key et al., 2022; Vikram et al., 2023), but primarily for final validation. Our work takes the critical next step by investigating *how to leverage property violations as actionable feedback for automated refinement*. As our core contribution, we further analyze the form of this feedback, demonstrating that minimized counterexamples are essential for effective refinement.

**Feedback Minimization for Automated Program Refinement.** The principle of minimizing failure-inducing inputs, established by delta-debugging (Misherghi & Su, 2006), is crucial for automated refinement. Recent work (Yang et al., 2025) has begun to explore this principle for LLMs to reduce their cognitive load and mitigate issues like the "lost-in-the-middle" (Liu et al., 2024c) problem. Our work advances this by integrating minimization into a property-driven paradigm. Instead of minimizing inputs that cause I/O mismatches, PGS minimizes counterexamples that violate high-level semantic properties. This shift finds bugs without reference outputs and forces a foundational question: what does "minimal" truly mean for an LLM? We thus provide the first systematic comparison of minimization proxies, empirically establishing that input token count is the most direct and effective signal for guiding the model's reasoning.

## 3 PILOT STUDY: WHAT MAKES FEEDBACK EFFECTIVE?

Before presenting our full framework, we conduct a pilot study to empirically validate our central hypothesis that the quality of feedback, defined by its content and form, is more critical for successful code refinement than its quantity. This study is designed to isolate and examine these two fundamental dimensions by answering two questions: first, regarding **feedback content**, is property-oriented feedback, which conveys semantic rules, more effective than traditional I/O-based feedback? Second, concerning **feedback form**, is structurally minimal feedback, derived from the simplest possible counterexample, more effective at guiding an LLM's repair?

To ensure our findings are robust and generalizable, our evaluation spans a diverse set of benchmarks, ranging from foundational function-level tasks (HumanEval (Chen et al., 2021) and MBPP (Austin et al., 2021), with their rigorous EvalPlus (Liu et al., 2024a) versions), to complex competition-level problems (LiveCodeBench (Jain et al., 2025)), and finally to real-world software issues (SWE-Bench (Jimenez et al., 2024)). Our analysis is performed on three state-of-the-art

| Strategy | | pass@1 (avg. of 3 models) | | | | | | Token Cost |
|---|---|---|---|---|---|---|---|---|
| Feature | Statistic | LCB-Easy | LCB-Mid | LCB-Hard | HumanEval | MBPP | SWE-bench | Avg. Tokens |
| Baseline | - | 76.0 | 36.4 | 10.5 | 85.8 | 65.3 | 16.6 | 4.73k |
| Line Coverage | Max | +8.9 | +5.2 | +2.1 | +6.1 | +7.3 | +3.3 | 5.53k |
| | Median | +9.5 | +5.3 | +3.0 | +6.3 | +7.2 | +3.2 | 5.12k |
| | Min | +12.4 | +5.8 | +3.7 | **+7.1** | +7.5 | **+3.7** | 4.87k |
| Runtime | Max | +8.6 | +5.3 | +2.1 | +6.1 | +7.2 | +3.2 | 5.76k |
| | Median | +11.2 | +5.3 | +3.5 | +6.6 | +7.4 | +3.4 | 5.10k |
| | Min | +11.4 | +5.6 | +3.8 | +7.0 | **+7.6** | +3.6 | 4.76k |
| Token Count | Max | +9.1 | +5.2 | +2.2 | +6.1 | +7.2 | +3.2 | 5.68k |
| | Median | +10.2 | +5.6 | +3.3 | +6.5 | +7.4 | +3.4 | 5.11k |
| | **Min** | **+13.2** | **+6.0** | **+4.0** | +7.0 | **+7.6** | **+3.7** | **4.72k** |

Table 1: **Comparison of Different Feedback Minimization Strategies.** Results show the average pass@1 improvement over the baseline across three models for nine feedback selection strategies. The final column shows the average token cost per attempt on the DeepSeek-R1-Distilled-32B model. Token counts are calculated using each model's respective tokenizer.

open-source LLMs: DeepSeek-Coder-V2 (Zhu et al., 2024), Qwen2.5-Coder (Hui et al., 2024), and the powerful reasoning model DeepSeek-R1-Distilled-32B (Guo et al., 2025). Further details are provided in Appendix B.

## 3.1 FEEDBACK CONTENT

**Motivation and Design.** Our first investigation provides the foundational evidence for our property-oriented principle. We aim to answer a critical question: *Does framing a bug as a violation of a general semantic property, rather than a specific input-output (I/O) mismatch, improve an LLM's ability to correct it?* To isolate this variable, we design an one-shot code refinement experiment using the DeepSeek-R1-Distilled-32B model. For each program that initially fails a public test case, we create two distinct refinement scenarios:

- **Simple I/O Feedback.** The LLM is given the standard failing test case, including the input, the expected output, and the erroneous output it produced.
- **Property-Oriented Feedback.** The LLM is given feedback that reframes the exact same failure as a violation of a high-level program property.

We compare the pass@1 rates of these refinement attempts against the performance of the initial, uncorrected code ("Baseline").

**Observation: Property-Oriented Feedback Drives More Effective Refinement.** As illustrated in Figure 2, the results reveal a clear and consistent performance hierarchy across all benchmarks. Property-oriented feedback consistently outperforms simple I/O feedback, which in turn offers only a modest improvement over the unrefined baseline. This advantage is particularly pronounced on more challenging problems. For instance, on the difficult LiveCodeBench-Hard, standard I/O feedback provides a small boost, increasing pass@1 from a 28.1% baseline to 32.0%. However, by simply reframing the exact same error as a property violation, the pass@1 jumps to 36.2%, nearly doubling the performance gain. This finding sug-

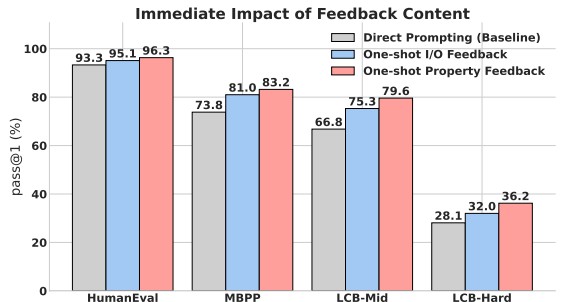

Figure 2: **The Impact of Feedback Content on One-shot Code Refinement.** We compare the pass@1 rates of the initial code against one-shot refinement using either standard I/O feedback or our proposed property-oriented feedback.

gests changing the feedback's content from a specific instance to a general rule allows the LLM to grasp the error's underlying semantic nature. This encourages the model to find a more generalizable solution that fixes the root cause, rather than merely patching the code to pass a single test case.

## 3.2 FEEDBACK FORM

**Motivation and Design.** Our second investigation provides the empirical evidence for our structurally minimal principle. While human programmers intuitively simplify failing test cases to isolate bugs (Misherghi & Su, 2006), this concept has not been systematically studied for LLM-based refinement. We investigate which proxy for "simplicity" is most effective at helping an LLM diagnose and fix a fault. Specifically, we create a pool of diverse counterexamples for a given bug and then select feedback based on the "min", "median", and "max" values of three distinct complexity metrics:

- **Line Coverage (Gopinath et al., 2014).** A classic software testing metric measuring the number of unique source code lines executed. Our hypothesis is that a test case with minimal line coverage isolates a simpler, more direct execution path to the fault.
- **Runtime.** A proxy for computational complexity, measured as execution time. We hypothesize that a shorter runtime corresponds to simpler program states (*e.g.*, fewer loop iterations), making the error easier to diagnose.
- **Input Token Count.** A direct measure of complexity from the LLM's perspective. The input for each counterexample is tokenized using the tokenizer specific to the LLM being used for refinement. We hypothesize this is the most effective metric because it directly minimizes the cognitive load on the model by presenting the most concise representation of the failure.

We evaluate these nine strategies and report the average pass@1 across all tested models.

**Observation: Minimization is Key, and Token Count is the Optimal Proxy.** The results, summarized in Table 1, reveal two decisive trends. First, across all three metrics, the minimization strategy ("min") consistently and significantly outperforms the "median" and "max" strategies. This provides strong evidence for the general principle of structural minimization: simpler is better. Second, when comparing the three minimization strategies, minimal "Token Count" consistently yields the highest pass@1 rates and is the most computationally efficient. While "Line Coverage" and "Runtime" are reasonable proxies for complexity, the number of input tokens is the most direct and reliable measure of the information an LLM must process. This analysis provides a clear guidance for our PGS framework's feedback design, *i.e.*, the most effective feedback is generated from the counterexample with the lowest input token count.

## 4 PROPERTY-GENERATED SOLVER

### 4.1 PROBLEM FORMULATION

The primary objective in the code generation task is to employ an LLM to generate a program $C$ based on a given natural language specification $Q$ and a set of public (visible) test cases $T_v$. Each test case $t_i = (I_i, O_i) \in T_v$ consists of an input $I_i$ and its corresponding expected output $O_i$. The quality of the generated program $C$ is ultimately evaluated against a set of private (hidden) test cases $T_h$. The program is considered correct if it passes all tests in $T_h$, *i.e.*, satisfying $\forall (I_j, O_j) \in T_h, C(I_j) = O_j$.

The core challenge in this problem is the generalization gap between the public tests $T_v$ and the private tests $T_h$. Consider that $T_v$ is often sparse, a generated program can easily overfit to the visible examples while still containing fundamental bugs that cause it to fail on $T_h$. While augmenting $T_v$ with auto-generated tests is a common remedy, it often compounds the issue by introducing significant feedback noise. Such auto-generated tests can inherit the model's own logical biases and struggle with the difficult test oracle problem, making them an unreliable signal for true correctness. Consequently, refinement approaches that rely on this limited and noisy pool of feedback are inherently capped in their effectiveness. The central task is therefore to generate feedback that transcends these limitations, guiding the LLM toward a truly generalizable and semantically robust solution.

### 4.2 FRAMEWORK OVERVIEW

As illustrated in Figure 3, we introduce the Property-Generated Solver (PGS), a framework that shifts the paradigm of TDD-based methods from relying on simple I/O feedback to utilizing property-oriented and structurally minimal feedback. PGS establishes a collaborative workflow

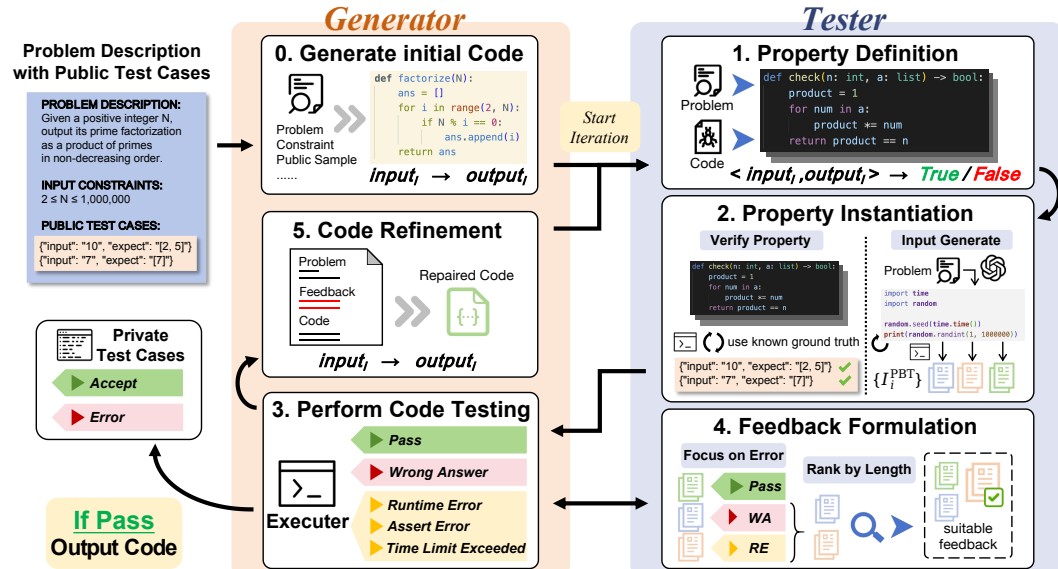

Figure 3: **Overview of the the Property-Generated Solver (PGS) Framework.** PGS establishes a collaborative workflow between the Generator and the Tester. **(0)** The Generator creates an initial program from the problem specification. **(1)** The Tester defines high-level properties and translates them into executable checks. **(2)** The Tester validates these properties and synthesizes a diverse pool of inputs for probing. **(3)** The program is executed, with property violations being identified. **(4)** The Tester analyzes property violations and formulates feedback from the minimal counterexample. **(5)** The Generator uses this targeted feedback to refine the code. The cycle (steps 1-5) then repeats.

between two specialized LLM-based agents, *i.e.*, a **Generator** and a **Tester**. Both agents can be implemented using general-purpose LLMs such as GPT-4 (OpenAI, 2023) or DeepSeek-R1 (Guo et al., 2025). The Generator performs code generation and refinement, while the Tester challenges the generated code by validating it against high-level properties and formulates minimal and highly actionable feedback based on property violation. The specific prompt templates used to instruct these agents are detailed in Appendix A.

**Initial Code Generation.** The process begins with the Generator producing an initial version of the program $C$ based on the problem specification $Q$. This serves as the starting point for the iterative refinement cycle.

**Property Definition.** Following the initial code generation, the Tester agent defines a set of candidate properties, $\mathcal{P}$, based on the natural language description $Q$. These properties are high-level semantic rules, ranging from global invariants (*e.g.*, "the output list must be sorted") to partial specifications (*e.g.*, "all elements in the output must be prime"). The Tester then translates these abstract properties into executable checking code, $C^{\mathcal{P}}$, typically structured as assertion statements or boolean-valued verification functions. To ensure the soundness of LLM-generated properties, we perform a rigorous validation step where each property check in $C^{\mathcal{P}}$ is validated against the public test cases $\boldsymbol{T}_v$. This critical filtering process discards any property that contradicts the known ground truth, ensuring that the feedback provided to the Generator is based on reliable and correct semantic rules.

**Input Synthesis for Property Probing.** Once a set of sound property checks $C^{\mathcal{P}}$ is established, the Tester generates a diverse pool of inputs, $\{I^{\mathcal{P}}\}$, to probe for violations. A key challenge is to create inputs that are semantically rich enough to stress-test the program's logic, a task where simple random or fuzzing techniques often prove insufficient as they tend to explore shallow logic flow. To overcome this, we adopt an LLM-driven synthesis technique (El-Kishky et al., 2025). We prompt a powerful LLM to act as a dedicated "test input generator," tasked with creating a rich and varied set of potential counterexamples based on the problem description and constraints. These synthesized inputs $\{I^{\mathcal{P}}\}$ are then used alongside the property checks $C^{\mathcal{P}}$ in the subsequent validation step, enabling PGS to uncover bugs that simple I/O checks would miss.

**Property-Driven Validation.** For validation, the Tester instruments the candidate program $C$ by injecting the property checks $C^{\mathcal{P}}$ directly into its source code, creating a unified executable $C'$. By treating properties as an intrinsic part of the program's logic, PGS transforms latent semantic bugs, which might otherwise result in a silent "Wrong Answer", into explicit, machine-checkable runtime errors like an *AssertionError*. This instrumentation is crucial, since it makes previously hidden bugs tractable, and further encourages the LLM to reason about the code and the problem specifications holistically. The instrumented program $C'$ is then executed against the full suite of inputs, including public test cases $\boldsymbol{T}_v$ and the synthesized inputs $\{I^{\mathcal{P}}\}$, to collect all outcomes for the next stage.

**Feedback Formulation.** A buggy program can fail on numerous inputs, but providing all raw outcomes would create a noisy and overwhelming signal. Inspired by delta-debugging principles (Misherghi & Su, 2006) and our pilot study, the Tester's role is to distill this information into a single, potent piece of feedback. From the set of all property-violating inputs, the Tester strategically selects the counterexample with the minimal input token count (a strategy empirically validated in Section 3.2). This structurally minimal feedback isolates the error's root cause, removes distracting noise, and provides the clear, actionable signal necessary for effective LLM-driven refinement.

**Code Refinement.** The Generator receives the property-oriented and structurally minimal feedback, which comprises the failing input, the program's erroneous output, and the specific violated property. This structured information is used to construct a new prompt that instructs the Generator to analyze the failure and produce a corrected version of the code. The iterative refinement cycle continues until the program passes all property checks and public test cases, or until a predefined budget is exhausted (*e.g.*, a maximum of five refinement attempts), steering the LLM toward a more robust and functionally correct solution.

## 5 EXPERIMENT

Our evaluation is designed to comprehensively assess the performance of the PGS framework. We first present the performance of PGS against several state-of-the-art baselines. We then conduct an in-depth analysis to dissect the core mechanisms that drive its effectiveness.

### 5.1 EXPERIMENTAL SETUP

**Comparison Counterparts.** We evaluate PGS against multiple state-of-the-art counterparts in code refinement, including: (1) non-iterative methods (Direct and CoT prompting (Wei et al., 2022)); (2) TDD frameworks that explicitly use test cases, such as Code-T (Chen et al., 2023a) and Reflexion (Shinn et al., 2024); and (3) sophisticated self-correction frameworks that emulate debugging, including Self-Edit (Zhang et al., 2023), Self-Debugger (Chen et al., 2023b), MGDebugger (Shi et al., 2024), and LDB (Zhong et al., 2024). To ensure a fair and direct comparison, all iterative methods were executed under a matched computational budget, capped at 5 refinement attempts, identical to PGS.

**Benchmarks and Foundation Models.** To assess the generalizability of our approach, we conduct experiments on four widely-recognized benchmarks that span a wide spectrum of tasks from function-level synthesis to competitive programming: HumanEval (Chen et al., 2021), MBPP (Austin et al., 2021), LiveCodeBench (Jain et al., 2025), and CodeContests (Li et al., 2022). We employ three powerful open-source models, listed in ascending order of their coding capabilities: DeepSeek-Coder-V2 (Zhu et al., 2024), Qwen2.5-Coder (Hui et al., 2024), and DeepSeek-R1-Distilled-32B (Guo et al., 2025).

**Implementation Details.** Our PGS framework runs for a maximum of 5 refinement iterations. In each iteration, we synthesize 5 candidate properties and generate 64 new test inputs using an LLM-created generator. All generation tasks use a temperature of 0.7. Guided by our pilot study (Section 3), our feedback strategy uses the property violation from the counterexample with the minimum number of input tokens, calculated using each model's respective tokenizer. To ensure statistically robust results, we report the average pass@1 score across 5 independent runs for each problem, thereby accounting for the stochasticity of the generation process. Further details are available in Appendix B.

Table 2: Comparison on Code Generation across Multiple Benchmarks. We report pass@1 scores with standard deviations. "DS" denotes DeepSeek, and "Claude-4" denotes Claude-sonnet-4. Fix Rate is calculated as the percentage of problems solved by PGS relative to the failure cases of the Baseline. Empty cells (−) indicate results omitted due to prohibitive cost or incompatibility. The best result in each row is highlighted in **bold**. Results marked with † are cited from original papers; others are reproduced in our experiments with standard deviations reported.

| Method | DS-V2 | Qwen2.5 | DS-R1-32B | Qwen3-30B | DS-V3.1 | Claude-4 |
|---|---|---|---|---|---|---|
| *HumanEval (HE)* | | | | | | |
| Baseline | $76.2 \pm 0.7$ | $87.8 \pm 0.6$ | $93.3 \pm 0.5$ | $91.5 \pm 0.8$ | $95.5 \pm 0.7$ | $97.2 \pm 0.6$ |
| CoT | $76.8 \pm 0.5$ | $87.8 \pm 0.5$ | $93.3 \pm 0.4$ | $91.5 \pm 0.7$ | $95.6 \pm 0.6$ | $97.4 \pm 0.5$ |
| Code-T | $81.1 \pm 1.7$ | $88.4 \pm 1.5$ | $94.5 \pm 1.4$ | $92.2 \pm 0.6$ | $96.2 \pm 0.5$ | $97.8 \pm 0.4$ |
| LDB† | 82.3 | - | - | - | - | - |
| Self-Edit | $81.7 \pm 1.6$ | $90.2 \pm 1.7$ | $95.1 \pm 1.6$ | $92.8 \pm 0.7$ | $96.5 \pm 0.6$ | $97.9 \pm 0.5$ |
| MGDebugger | $83.5 \pm 1.7$ | $92.1 \pm 1.6$ | $95.7 \pm 1.5$ | $93.2 \pm 0.8$ | $96.8 \pm 0.7$ | $98.2 \pm 0.6$ |
| Self-Debug | $84.1 \pm 1.9$ | $92.7 \pm 1.8$ | $96.3 \pm 1.7$ | $93.5 \pm 0.9$ | $97.1 \pm 0.8$ | $98.5 \pm 0.7$ |
| Reflexion | $86.6 \pm 1.2$ | $91.5 \pm 1.4$ | $95.1 \pm 1.3$ | $92.9 \pm 0.6$ | $96.9 \pm 0.8$ | $98.5 \pm 0.6$ |
| **PGS (Ours)** | $\mathbf{89.0 \pm 1.5}$ | $\mathbf{94.5 \pm 1.1}$ | $\mathbf{97.6 \pm 1.0}$ | $\mathbf{95.2 \pm 0.8}$ | $\mathbf{98.2 \pm 0.6}$ | $\mathbf{99.1 \pm 0.3}$ |
| *Fix Rate* | *53.8%* | *54.9%* | *64.2%* | *43.5%* | *60.0%* | *67.9%* |
| *MBPP* | | | | | | |
| Baseline | $56.8 \pm 0.6$ | $65.4 \pm 0.7$ | $73.8 \pm 0.6$ | $73.1 \pm 0.9$ | $89.5 \pm 0.8$ | $93.5 \pm 0.3$ |
| CoT | $57.2 \pm 0.4$ | $66.6 \pm 0.5$ | $73.8 \pm 0.5$ | $73.1 \pm 0.8$ | $89.8 \pm 0.7$ | $93.8 \pm 0.3$ |
| Code-T | $60.4 \pm 1.5$ | $69.4 \pm 1.6$ | $82.4 \pm 1.0$ | $76.5 \pm 1.2$ | $90.8 \pm 1.1$ | $94.5 \pm 0.5$ |
| LDB† | 62.6 | - | - | - | - | - |
| Self-Edit | $62.4 \pm 1.5$ | $70.2 \pm 1.8$ | $83.0 \pm 1.2$ | $78.2 \pm 1.3$ | $91.2 \pm 1.2$ | $94.8 \pm 0.6$ |
| MGDebugger | $63.8 \pm 2.0$ | $71.2 \pm 1.9$ | $83.8 \pm 1.3$ | $79.6 \pm 1.4$ | $91.5 \pm 1.3$ | $95.2 \pm 0.7$ |
| Self-Debug | $63.8 \pm 1.9$ | $72.4 \pm 2.0$ | $84.4 \pm 1.4$ | $80.0 \pm 1.5$ | $91.8 \pm 1.4$ | $95.5 \pm 0.8$ |
| **PGS (Ours)** | $\mathbf{67.6 \pm 1.8}$ | $\mathbf{76.6 \pm 1.9}$ | $\mathbf{87.2 \pm 1.3}$ | $\mathbf{82.5 \pm 1.5}$ | $\mathbf{94.1 \pm 1.4}$ | $\mathbf{96.5 \pm 0.8}$ |
| *Fix Rate* | *25.0%* | *32.4%* | *51.1%* | *34.9%* | *43.8%* | *46.2%* |
| *LiveCodeBench (LCB)* | | | | | | |
| Baseline | $26.7 \pm 0.8$ | $31.8 \pm 0.9$ | $64.4 \pm 0.9$ | $52.2 \pm 1.0$ | $72.5 \pm 2.6$ | $63.1 \pm 1.7$ |
| CoT | $26.9 \pm 0.9$ | $32.4 \pm 0.8$ | $64.4 \pm 0.8$ | $52.2 \pm 0.9$ | $72.7 \pm 2.2$ | $63.2 \pm 1.6$ |
| Code-T | $29.2 \pm 1.3$ | $34.6 \pm 1.4$ | $70.8 \pm 1.6$ | $54.5 \pm 1.8$ | $75.5 \pm 2.7$ | $68.2 \pm 1.4$ |
| Self-Edit | $30.2 \pm 1.9$ | $35.2 \pm 1.8$ | $73.6 \pm 1.8$ | $60.2 \pm 2.0$ | $79.8 \pm 2.8$ | $70.5 \pm 1.9$ |
| Self-Debug | $31.3 \pm 2.2$ | $38.5 \pm 2.0$ | $72.5 \pm 2.0$ | $61.5 \pm 2.1$ | $80.5 \pm 2.0$ | $72.8 \pm 2.0$ |
| **PGS (Ours)** | $\mathbf{34.1 \pm 1.7}$ | $\mathbf{40.0 \pm 1.9}$ | $\mathbf{76.5 \pm 1.8}$ | $\mathbf{65.1 \pm 1.4}$ | $\mathbf{83.2 \pm 2.0}$ | $\mathbf{75.5 \pm 1.8}$ |
| *Fix Rate* | *10.1%* | *12.0%* | *34.0%* | *27.0%* | *38.9%* | *33.6%* |
| *CodeContest (CC)* | | | | | | |
| Baseline | $12.5 \pm 0.6$ | $14.4 \pm 0.7$ | $38.1 \pm 0.8$ | $30.8 \pm 0.9$ | $46.8 \pm 1.1$ | $42.1 \pm 1.0$ |
| CoT | $12.8 \pm 0.8$ | $14.9 \pm 0.8$ | $38.1 \pm 0.8$ | $30.8 \pm 0.7$ | $47.2 \pm 1.0$ | $42.6 \pm 0.9$ |
| Code-T | $14.2 \pm 0.9$ | $15.9 \pm 1.0$ | $42.9 \pm 1.4$ | $33.2 \pm 1.6$ | $49.6 \pm 1.4$ | $44.3 \pm 1.5$ |
| Self-Edit | $15.6 \pm 1.7$ | $16.4 \pm 1.5$ | $44.8 \pm 1.6$ | $34.4 \pm 1.7$ | $53.2 \pm 1.6$ | $48.5 \pm 1.6$ |
| Self-Debug | $16.1 \pm 2.1$ | $17.3 \pm 1.9$ | $45.8 \pm 1.9$ | $36.1 \pm 2.0$ | $54.5 \pm 1.8$ | $49.9 \pm 1.9$ |
| **PGS (Ours)** | $\mathbf{20.2 \pm 1.8}$ | $\mathbf{22.4 \pm 2.0}$ | $\mathbf{49.7 \pm 2.0}$ | $\mathbf{41.7 \pm 2.2}$ | $\mathbf{60.2 \pm 2.1}$ | $\mathbf{55.9 \pm 2.1}$ |
| *Fix Rate* | *8.8%* | *9.3%* | *18.7%* | *15.8%* | *25.2%* | *23.8%* |
| *SWE-bench* | | | | | | |
| SWE-agent | $9.8 \pm 1.4$ | $10.3 \pm 0.8$ | $34.4 \pm 2.0$ | $46.5 \pm 1.7$ | $54.2 \pm 1.8$ | $65.5 \pm 1.6$ |
| **PGS (Ours)** | $\mathbf{11.9 \pm 1.0}$ | $\mathbf{12.8 \pm 0.5}$ | $\mathbf{37.3 \pm 2.3}$ | $\mathbf{50.7 \pm 1.9}$ | $\mathbf{58.4 \pm 2.5}$ | $\mathbf{70.2 \pm 1.5}$ |

## 5.2 MAIN RESULT

**Overall Performance.** Grounded in our empirically validated principles of effective feedback, the PGS framework consistently establishes a new state-of-the-art for code refinement. As shown in Table 2, PGS substantially outperforms all baseline methods across every benchmark and all three language models. This demonstrates the model-agnostic benefits of our high-quality feedback approach. This advantage is particularly pronounced on the challenging LiveCodeBench benchmark,

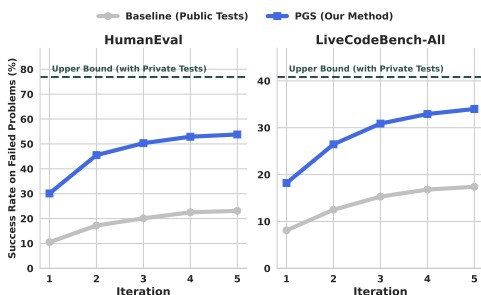 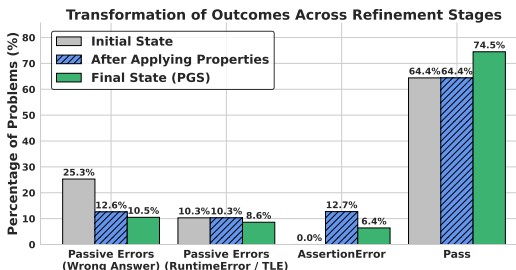

Figure 4: Iterative fix rate of PGS vs. a baseline on initially-failed problems from **HumanEval** (with Deepseek-Coder-V2) and **LiveCodeBench** (with DeepSeek-R1-Distilled-32B). The dashed line indicates the performance upper bound.

Figure 5: The core PGS mechanism: transformation of outcomes across refinement stages. Ambiguous "Wrong Answer" failures are converted into explicit "AssertionError"s, which are then resolved, increasing the final "Pass" rate.

where PGS with DeepSeek-R1-Distilled-32B achieves a 12.1% improvement over the baseline, far surpassing other iterative methods.

**Comparison with State-of-the-Art Baselines.** PGS outperforms sophisticated debugging frameworks. On HumanEval, for instance, PGS with DeepSeek-Coder-V2 reaches 89.0% pass@1, a significant lead over advanced methods like MGDebugger (83.5%) and LDB (82.3%). While these methods excel at tracing execution for given tests, their scope is fundamentally limited by those initial tests. PGS, by contrast, actively probes the code's semantic boundaries, allowing it to uncover and repair a wider class of bugs.

### 5.3 ABLATION STUDY

#### 5.3.1 EFFECTIVENESS OF INPUT SYNTHESIS

A core component of PGS framework is the Tester agent's ability to synthesize new inputs to probe for property violations. To rigorously quantify the generated value, we focus on the most challenging subset of problems, *i.e.*, those where the LLM's initial code generation failed the hidden tests. On this difficult subset, we compare the our PGS framework against a baseline TDD approach. We also establish a practical performance ceiling, representing the maximum possible fix rate if the model had access to feedback from all hidden private tests.

The results, as shown in Figure 4, reveal the decisive advantage of our synthesis-driven approach. On both HumanEval and LiveCodeBench, PGS not only starts with a higher fix rate but also achieves its gains more rapidly than the baseline, with the performance gap widening at each iteration. By the final iteration on **HumanEval**, the baseline TDD approach plateaus, solving only **23.1%** of the initially failed problems. In stark contrast, PGS reaches a much higher success rate of **53.8%**. Furthermore, the practical performance ceiling (76.9%) places our method's performance in a powerful context. This means that PGS successfully recovers over 57% of the total performance gap between the baseline and the oracle. The trend is consistent on the more challenging **LiveCodeBench-All** dataset, where PGS closes an impressive 71% of the performance gap between the baseline (17.4%) and the oracle (40.8%). This proves that the inputs synthesized by our Tester are not redundant. They are crucial for uncovering the same kinds of deep logical flaws that are typically only caught by hidden private tests, validating the effectiveness of our overall strategy.

#### 5.3.2 COST-EFFECTIVENESS ANALYSIS

We provide a cost-effectiveness analysis measuring the trade-off between accumulated token consumption and Pass@1 performance across 5 refinement iterations. When compared with simple baselines like Self-Edit, although PGS consumes more tokens (1.6×) per iteration due to property generation step, this cost is quickly offset by its superior and performance gains. As illustrated in Fig 6, PGS after only 2 iterations already surpasses the final performance of the Self-Edit after 5 full iterations with fewer token costs on both HumanEval and LiveCodeBench. When compared with

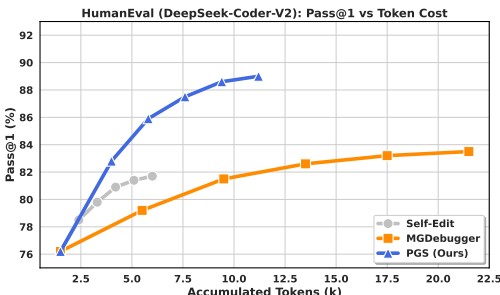 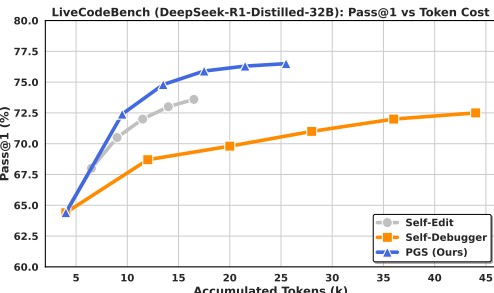

Figure 6: **Pass@1 Performance *vs.* Token Consumption.** Token counts are calculated using each model's respective tokenizer.

complex step-by-step debugging baselines like Self-Debugger, PGS is significantly more efficient than these methods, both on Pass@1 and token costs. Self-Debugger requires nearly 4 times more tokens to achieve the same performance as PGS. In summary, PGS offers the optimal trade-off, delivering the highest accuracy with lower computational overhead than its counterparts.

### 5.3.3 TRANSFORMING LATENT BUGS INTO ACTIONABLE SIGNALS

Having established the effectiveness of our synthesized inputs, we now analyze the core mechanism that makes the resulting feedback so potent. Our framework's key design choice is to have the Generator integrate the property-checking code directly into the program being tested. We hypothesize that this transforms ambiguous failures into explicit, actionable ones. To verify this, we visualize the distribution of failure types at three key stages of the process, as detailed in Figure 5: **(1)** the **Initial** state of the code generated by baseline; **(2)** the state **After Applying Properties**, where the property-checking code is injected into the initial code; and **(3)** the **Final** state after PGS completes its refinement.

The transition from Stage 1 to Stage 2 vividly reveals the core of the mechanism. Initially, "Wrong Answer" is a dominant failure mode, accounting for **25.3%** of outcomes. After simply instrumenting the code with properties (Stage 2), with the "Pass" rate holding steady at 64.4%, this category is nearly halved to **12.6%**. Critically, this reduction is almost perfectly mirrored by the emergence of the "AssertionError" category. This is not a change in the code's logic, but a change in its observability. This vividly illustrates that the injected properties act as a diagnostic lens, successfully **surfacing latent bugs by converting them from vague semantic errors into explicit, machine-checkable violations**.

This transformation is critical for the final stage. Armed with this clear "AssertionError" signal, the Generator effectively debugs the code. In the final state (Stage 3), the "AssertionError" category is reduced to just **6.4%**, and other passive errors also decrease. This successful resolution directly translates into a significant increase in the final 'Pass' rate, which rises from 64.4% to **74.5%**. This analysis confirms that PGS operates on a clear two-step principle: first, it makes hidden bugs visible via property injection, and second, it leverages these clear signals for effective refinement. A detailed case study tracing this process is available in the Appendix C.

## 6 CONCLUSION

We tackle the challenge of improving LLM-generated code correctness by establishing a foundational principle: Effective refinement is driven by feedback that is both property-oriented and structurally minimal. This core insight provides a new lens through which to understand and build automated refinement systems. We introduce PGS, a PBT-based framework designed to operationalize this principle. Our extensive experiments show that PGS achieves state-of-the-art performance on rigorous benchmarks by systematically converting latent logical bugs into explicit, minimal refinement signals. By focusing on feedback quality over test quantity, our work provides a robust methodology for more reliable code generation. A key direction for future work is enhancing autonomous property discovery.

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

# A  PROPERTY-GENERATED SOLVER SETTINGS

## A.1  CATEGORY OF CODE EXECUTION OUTCOMES

In the main body of the paper, we discussed that our framework's effectiveness stems from generating high-quality, targeted feedback. A cornerstone of this strategy is the ability to differentiate between various outcomes of code execution. Each outcome category provides a distinct type of signal about the nature of the flaw in the generated code, allowing our Tester agent to formulate the most effective and actionable feedback for the Generator agent. Below, we detail these categories and the corresponding feedback strategy employed by PGS.

**Pass.** This is the successful outcome, where the candidate code correctly passes a given test case (either a public test or a property-driven check). No corrective feedback is needed for this instance. If the code passes all available public tests and all synthesized property checks, the refinement process concludes successfully.

**Property Violation (`AssertionError`).** This is the primary signal leveraged by the PGS framework and represents the most informative type of failure. A property violation occurs when the code executes without crashing but fails a custom-injected assertion that encodes a high-level program invariant (e.g., a sorted list must be non-decreasing). This mechanism is crucial for converting *latent logical flaws*—bugs that might not be caught by simple input-output checks—into explicit, machine-checkable failures. **Feedback Strategy:** This outcome receives the highest priority. The Tester provides feedback centered on the minimal counterexample that triggered the `AssertionError`, along with a description of the violated property. This gives the Generator a clear, semantic-level clue about what is wrong with its logic.

**Wrong Answer (WA).** This outcome occurs when the program's output does not match the expected output for a known, ground-truth test case (typically the public examples provided with the problem). This indicates a fundamental logical error. **Feedback Strategy:** The feedback consists of the input, the model's incorrect output, and the correct, expected output. This is the classic feedback loop in Test-Driven Development (TDD), and PGS uses it to correct basic functional errors.

**Runtime Error (RE).** This category includes errors that cause the program to crash during execution, such as `IndexError`, `TypeError`, or `ZeroDivisionError`. These errors often point to faulty assumptions about the input data or incorrect state management. **Feedback Strategy:** The feedback includes the full error message and stack trace, along with the specific input that caused the crash. This provides the Generator with a precise pointer to the location and type of the bug in the code.

**Time Limit Exceeded (TLE).** This outcome signifies that the generated code is correct in its logic but is too slow to pass within the allocated time constraints. This points to a flaw in algorithmic efficiency rather than logical correctness. **Feedback Strategy:** The feedback highlights the inefficiency. It provides the input that caused the TLE and explicitly prompts the Generator to rethink its algorithmic approach to find a more time-efficient solution. This guides the model towards high-level algorithmic optimization.

## A.2 PROMPT TEMPLATES

This section provides the detailed prompt templates used throughout our Property-Generated Solver (PGS) framework. These prompts are designed to guide the Generator and Tester agents through the key stages of the code refinement process. The prompts are designed to be clear and role-specific, ensuring that each agent performs its designated task effectively.

Each prompt is tailored for a specific stage of the refinement process, guiding our agents to perform their tasks. The process begins with the **Initial Code Generation** prompt, which instructs the Generator to produce a baseline solution (Table 6). Concurrently, the Tester agent uses the **Property Generation** prompt to distill high-level program properties (Table 7). Subsequently, the Tester employs the **Input Synthesis** prompt to generate diverse inputs for probing these properties (Table 8). Finally, if a violation is detected, the cycle culminates in the **Feedback-Driven Repair** prompt, which provides the Generator with a minimal counterexample to fix the bug (Table 9).

## B    MORE EXPERIMENTAL SETUP

This section details the experimental setup, including the hardware and software environment, as well as the specific parameters used for model inference during our experiments.

**Benchmarks.** Following prior works Zhang et al. (2024); Shi et al. (2024); Chen et al. (2023b), our evaluation utilizes three prominent code generation benchmarks:

- **HumanEval Chen et al. (2021).** A standard benchmark comprising 164 handwritten Python programming problems designed to evaluate the function-level code synthesis capabilities of LLMs. During the generation and refinement process, models are provided with the problem description and any canonical tests accompanying the original HumanEval problem statements. Final validation is performed using the benchmark's standard hidden test cases.

- **MBPP Austin et al. (2021).** This benchmark consists of approximately 500 crowd-sourced entry-level Python programming problems. The models receive the problem description and the first hidden test case during the generation phase Ni et al. (2023). Final validation is performed using the benchmark's standard hidden test cases.

- **LiveCodeBench Jain et al. (2025).** A challenging benchmark featuring problems sourced from live programming contests, often requiring more complex algorithmic reasoning, intricate I/O handling, and adherence to stricter execution constraints. To ensure a comprehensive and up-to-date evaluation, we utilize the latest "v5" version, comprising 880 problems. For all problems from this benchmark, the public test cases provided with each problem description are made available to all Test-Driven Development methods, including PGS and relevant baselines.

- **CodeContests Li et al. (2022).** A highly competitive benchmark that assesses advanced problem-solving and algorithmic implementation skills. While the full dataset features thousands of problems from various competitions, our evaluation utilizes the official `valid` split, which consists of 117 problems. During the generation and refinement process, models are given access to the problem description along with the public (sample) test cases. Final validation is performed against a comprehensive set of hidden test cases, demanding solutions that are not only correct but also efficient enough to pass strict time and memory limits.

- **SWE-Bench Jimenez et al. (2024).** A rigorous and realistic benchmark designed to evaluate LLMs on genuine software engineering tasks. Our evaluation focuses on the challenging "Verified" split, which comprises 500 real-world bug-fixing scenarios extracted from popular GitHub repositories. For each task, models are provided with the full code repository, the problematic GitHub issue description, and any associated pull request information. Final validation is performed by executing the project's original test suite against the generated patch, requiring the fix to resolve the issue without introducing any regressions.

**Comparison Baselines.** We compare PGS against a carefully selected suite of baselines that represent the key paradigms in state-of-the-art code refinement. We categorize them as follows:

- **Foundational Baselines.** These methods measure the raw capabilities of the LLMs.
  - *Direct & CoT Prompting.* We evaluate both zero-shot baseline and Chain-of-Thought (CoT) (Wei et al., 2022) prompting to establish the fundamental performance ceiling without iterative refinement.

- **Test-Driven Refinement Frameworks.** These methods explicitly use test outcomes to guide the refinement process, representing the direct alternative to our approach.
  - *Code-T* (Chen et al., 2023a). Leverages auto-generated tests to rank code candidates.
  - *Reflexion* (Shinn et al., 2024). A method that uses self-reflection to improve code.

- **Self-Correction and Debugging Frameworks.** These are more sophisticated methods that emulate a debugging process to identify and fix errors.
  - *Self-Edit* (Zhang et al., 2023). A technique where the LLM attempts to refine its own code based on execution feedback.
  - *Self-Debugger* (Chen et al., 2023b). An iterative method where the LLM explains its code and simulates a debugging process to fix bugs.
  - *MGDebugger* (Shi et al., 2024). A multi-level framework that identifies and fixes errors at different levels of code abstraction.
  - *LDB* (Zhong et al., 2024). A refinement technique that tracks intermediate variable values during runtime to locate and repair errors.
  - *SWE-agent* Yang et al. (2024): An open source framework for repository-level code refinement, which uses a ReAct loop with specialized tools for navigating and editing codebases. We include it as a primary baseline for SWE-bench experiments.

All methods are provided with identical problem descriptions and public test cases to ensure a fair comparison. Private test cases are strictly withheld during the iterative loops and are used solely for final evaluation.

**Foundation Models.** We select three LLMs with different capabilities to implement proposed PGS. Based on their general coding proficiency, they are listed from weak to strong as follows:

- **DeepSeek-Coder-V2 Zhu et al. (2024).** A powerful open-source model specifically optimized for code generation tasks.

- **Qwen2.5-Coder Hui et al. (2024).** A strong open-source model from the Qwen series, known for its advanced coding abilities.

- **DeepSeek-R1-Distilled-32B Guo et al. (2025).** A highly capable LLM featured with long CoT reasoning. We utilize a variant 32B distilled model, which aims to offer a strong balance of performance and efficiency.

For all models, we follow official configurations (*e.g.*, maximum context window of tokens, temperature, specific version identifiers) to guarantee a consistent setup.

**Metrics.** We adopt two metrics to evaluate the effectiveness of PGS:

- **pass@1 Yu et al. (2024)** measures the overall proportion of problems for which the generated final code successfully passes all hidden (private) test cases.

- **Fix Rate(Repair Success Rate) Yasunaga & Liang (2021)** quantifies the proportion of initially incorrect code samples that are successfully corrected by the iterative refinement process to pass all hidden test cases.

**Tool Usage and Grounding.** PGS employs different tooling strategies based on task complexity.

- **Function-level tasks (e.g., HumanEval, MBPP)**: We use lightweight execution environments for Python code validation.

- **Competition-level tasks (e.g., LiveCodeBench, CodeContests)**: We incorporate problem specific I/O handlers and time constraints.

- **Repository-level tasks (e.g., SWE-bench)**: To ensure fair comparison, we adopt the same toolset and grounding as SWE-agent, including bash terminal, file navigation, edit commands, and test execution.

**Hardware and Software Environment.** All experiments were conducted on a server equipped with **4x NVIDIA H100 GPUs**. The operating system environment was configured with **Python 3.11** and **CUDA 12.2**. Our software stack was built upon **PyTorch 2.6**, and we utilized the **vLLM library (version 0.8.5.post1)** for efficient and high-throughput inference of all large language models. To maximize performance, we configured vLLM to use a **4-way tensor parallelism** strategy across the available GPUs.

**Inference Parameters.** For all code generation and refinement tasks, we employed a consistent sampling strategy to ensure comparability across models and methods. The maximum number of tokens for a single generation was set to `max_tokens=32768`, providing ample capacity for the models to generate complete solutions and detailed reasoning steps without premature truncation. For sampling, we used a nucleus sampling approach with a `top_p` value of **0.95**. This choice is informed by the technical report for models like DeepSeek-R1 and Qwen-2.5-Coder, which suggests this setting promotes a good balance between creativity and factual coherence in generation tasks.

**Baseline Consistency.** To ensure a fair and rigorous comparison, all baseline methods evaluated in our study were executed using the identical hardware setup and inference configuration described above. Specifically, they shared the same sampling strategy (`top_p=0.95`) and `max_tokens` setting. This consistency eliminates variability due to different generation parameters and ensures that observed performance differences can be attributed to the core methodologies of the frameworks themselves.

## C CASE STUDIES

To provide a concrete, step-by-step illustration of our Property-Generated Solver (PGS) framework in action, we present two detailed case studies. These cases are chosen to demonstrate how the core principles discussed in the main body—property-oriented feedback and structural minimization—are operationalized to resolve different types of challenging bugs. Each case follows the full refinement workflow, from a flawed initial code to a correct final version, highlighting how targeted feedback drives effective debugging.

The first case study, presented in Figure 7, showcases our framework's primary mechanism for **transforming latent logical flaws into explicit, actionable signals**. It begins with an initial solution that is overly complex and contains a subtle bug related to unhandled states (resulting in an infinite cost). We show how injecting simple, property-based assertions (e.g., 'cost must be finite') allows a minimal counterexample ("0011") to trigger a clear `AssertionError`. This precise, low-noise signal pinpoints the exact logical failure, guiding the LLM to produce a final solution that is not only correct but also significantly more concise and elegant.

The second case study, detailed in Figure 8, demonstrates PGS's ability to **invalidate an entire algorithmic approach by using a simpler, trusted implementation as a property**. Here, the initial code attempts a sophisticated optimization using cycle detection, which contains a flaw. The key property injected is that for a subset of simple inputs (small 'k'), the output of the complex algorithm must match that of a direct, brute-force simulation. The simulation acts as a temporary "oracle." The resulting `AssertionError` explicitly shows a discrepancy between the optimization and the trusted simulation. This potent feedback prompts the model to abandon the flawed approach entirely and converge on a robust, correct solution using a different paradigm (dynamic programming with binary lifting), a feat difficult to achieve with traditional feedback.

## D MORE ABLATION STUDY

In this section, we provide a deeper analysis of the underlying mechanisms of PGS, dissecting the verification asymmetry, the contribution of individual components, and the impact of dataset versions on evaluation.

Table 3: **Asymmetry of Verification.** We evaluate DeepSeek-R1-Distilled-32B's generation and verification accuracy on 100-problem subset of LiveCodeBench. Here, "GenAcc." denotes the standard pass@1 rate of generating the correct solution code. "VerAcc." denotes the accuracy of the model in generating a valid property that correctly distinguishes correct from incorrect solutions. "w/ filtering" denotes using public test cases to filter out invalid properties.

| Difficulty | GenAcc. | VerAcc. | VerAcc. (w/ filtering) |
|---|---|---|---|
| Easy ($n = 32$) | 90.6% (29/32) | 93.8% (30/32) | 96.9% (31/32) |
| Medium ($n = 34$) | 67.6% (23/34) | 91.2% (31/34) | 94.1% (32/34) |
| Hard ($n = 34$) | 32.4% (11/34) | 76.5% (26/34) | 88.2% (30/34) |
| **Overall** ($N = 100$) | **63.0%** (63/100) | **87.0%** (87/100) | **93.0%** (93/100) |

## D.1 ASYMMETRY OF VERIFICATION

"Cycle of self-deception" is the fundamental challenge in LLM-based self-correction, where the LLM shares the same logical biases when it generates the code or predicts an oracle. Our PGS framework is designed to break this cycle by leveraging a core principle: the **Asymmetry of Verification** (Wei, 2025), *i.e.*, verifying a solution's correctness is often a significantly easier task than generating that solution from scratch.

**Roles of Generator and Tester in PGS.** The core design of PGS adheres to this principle. The Generator in PGS is responsible for producing complex algorithmic logic (*e.g.*, a complete prime factorization algorithm), while the Tester only needs to generate several simple, independent property checks (*e.g.*, `assert math.prod(factors) == N`). The risk of an LLM failing to generate a simple assertion (verification) is significantly lower than its risk of failing to generate a complex algorithm (generation), thus ensuring that the Tester can provide reliable and valid guidance even when the Generator fails.

**Empirical Evidence.** To provide rigorous empirical evidence for this asymmetry, we conduct an additional experiment on a 100-problem subset of LiveCodeBench, which consists of 32 Easy, 34 Medium, and 34 Hard problems, using DeepSeek-R1-Distilled-32B. Specifically, we compare the LLM's success rate at generation (*e.g.*, solving the problem from scratch) versus its success rate at verification (*e.g.*, writing a verifier that correctly identifies correct and incorrect solutions). The results are present in Table 3. Our results empirically confirm this asymmetry of verification. The accuracy of verification (87.0%) is significantly higher than that of code generation (63.0%). Especially on hard problems, the verification accuracy (76.5%) is more than twice the generation accuracy (32.4%). Notably, when applying property validation mentioned in Section 4.2 (w/ filtering), it boosts the verification accuracy on Hard problems from 76.5% to 88.2%, further widening the generation-to-verification gap. This ensures the property-based feedback provided to the Generator is reliable and precise.

## D.2 CONTRIBUTION OF EACH COMPONENT

To investigate the contribution of each component within PGS (*e.g.*, structural minimization, property generation & validation, iterative refinement), we conduct an ablation study on LiveCodeBench using DeepSeek-R1-Distilled-32B. The results are present in Table 4.

**Impact of Structural Minimization (+2.8%).** Simply switching the feedback mechanism from reporting a random failing input to reporting the minimal failing input (based on token count) improves the pass rate from 64.4% to 67.2%. This validates our hypothesis that even in the absence of property, reducing the cognitive load of the feedback signals effectively enhances the model's debugging capability.

**Impact of Property Generation (+1.3%) & Validation (+3.1%).** Transitioning from I/O-based feedback to Property-Oriented feedback yields a modest gain. This suggests that the quality of property generated by LLM may vary significantly, potentially containing hallucinations or logical flaws. Applying property validation mechanism provides a significant boost, indicating that using high-quality properties after filtering to generate feedback can effectively provide robust semantic guidance for subsequent refinement processes.

Table 4: **Contribution of Each Component in PGS.**

| Method / Component | Pass@1 (%) | Δ Improvement |
|---|---|---|
| **Baseline** (Raw I/O Feedback) | 64.4 | - |
| + Structural Minimization | 67.2 | +2.8 |
| + Property Generation | 68.5 | +1.3 |
| + Property Check (Filtering) | 71.6 | +3.1 |
| + Iterative Refinement (**Full PGS**) | **76.5** | +4.9 |

Table 5: **Multi-Lingual Evaluation on HumanEval-X and LiveCodeBench.** We test PGS's code refinement capability on C/C++ language. All results are reported using mean $\pm$ std from 5 runs.

| Method | DeepSeek-Coder-V2 | | Qwen2.5-Coder | | DeepSeek-R1-Distilled-32B | |
|---|---|---|---|---|---|---|
| | HumanEval-C | LCB-C | HumanEval-C | LCB-C | HumanEval-C | LCB-C |
| Baseline | $75.1 \pm 0.8$ | $25.2 \pm 0.9$ | $86.5 \pm 0.7$ | $30.1 \pm 1.0$ | $92.1 \pm 0.6$ | $62.5 \pm 1.0$ |
| CoT | $75.5 \pm 0.6$ | $25.5 \pm 0.8$ | $86.5 \pm 0.6$ | $30.5 \pm 0.9$ | $92.1 \pm 0.5$ | $62.5 \pm 0.9$ |
| Code-T | $79.5 \pm 1.5$ | $27.8 \pm 1.2$ | $87.2 \pm 1.4$ | $32.8 \pm 1.3$ | $93.5 \pm 1.3$ | $68.8 \pm 1.4$ |
| Self-Edit | $80.2 \pm 1.6$ | $28.5 \pm 1.5$ | $89.1 \pm 1.6$ | $33.5 \pm 1.6$ | $94.2 \pm 1.5$ | $71.5 \pm 1.7$ |
| Self-Debug | $82.5 \pm 1.8$ | $29.8 \pm 1.7$ | $91.5 \pm 1.8$ | $36.8 \pm 1.8$ | $95.5 \pm 1.7$ | $70.8 \pm 1.9$ |
| **PGS (Ours)** | $\textbf{87.8} \pm \textbf{1.6}$ | $\textbf{32.8} \pm \textbf{1.8}$ | $\textbf{93.5} \pm \textbf{1.6}$ | $\textbf{38.5} \pm \textbf{2.0}$ | $\textbf{96.8} \pm \textbf{1.5}$ | $\textbf{74.8} \pm \textbf{1.9}$ |

**Impact of Iterative Refinement (+4.9%).** Finally, allowing the LLM to engage in a multi-round refinement loop further improves performance to 76.5%. This confirms that complex bugs often require stepwise correction rather than a single-shot fix.

### D.3 MULTI-LINGUAL CODE GENERATION

To validate the generalizability of PGS beyond Python, We evaluated PGS on C++ versions of HumanEval (HumanEval-X (Zheng et al., 2023)) and LiveCodeBench, maintaining identical problem logic to their Python counterparts. We adopt 5 refinement iterations, 0.7 temperature and minimal token count strategy for each run. As shown in Table 5, PGS demonstrates consistent improvements across all models and benchmarks. On HumanEval-C/C++ with DeepSeek-R1-Distilled-32B, PGS achieves 96.8% pass@1, outperforming the baseline (92.1%) by 4.7 points. On the more challenging LCB-C/C++, PGS shows even larger gains (PGS 74.8% vs. baseline 62.5%). The fix rate remains robust, averaging 51.4% across C/C++ benchmarks. Thus, PGS maintains its effectiveness across C++ languages, validating its generalizability to diverse code generation scenarios.

## E LIMITATION AND FUTURE WORK

A primary limitation of our framework lies in its dependence on the LLM's ability to generate high-quality properties, a prerequisite for creating effective, property-oriented feedback. We recognize that for certain problems, formulating properties that are both correct and comprehensive can be an inherently difficult task. Another limitation is that even with precise, property-oriented feedback, the LLM may still fail to correct deeply rooted or complex logical faults, sometimes getting stuck in refinement loops. These challenges point to future work in enhancing autonomous property discovery. Furthermore, investigating multi-turn repair strategies, where the model can reflect on a history of failed attempts or decompose a complex bug into sub-problems, could significantly boost its problem-solving capabilities.

## F LLM USAGE STATEMENT

In the preparation of this manuscript, a large language model (LLM) was utilized as a writing assistant. Its role was strictly limited to an editorial and technical support capacity. Specific tasks included rephrasing sentences for better flow, correcting grammar, suggesting alternative phrasing, and providing advice on LaTeX formatting. The LLM was not used for research ideation, gener-

Table 6: **Prompt Templates for Code Generation.** This table illustrates how prompts are constructed. Placeholders like {question_content} and {starter_code} are replaced with actual content during runtime.

---

**1. Core User Prompt Template (Generated by `get_generic_question_template_answer`)**

This is the base structure for the user's request, with a key variation based on the presence of starter code.

**### Question:**
{question_content}

**IF** starter_code is provided:
**### Format:** You will use the following starter code to write the solution...
```python\n{starter_code}\n```
**ELSE**:
**### Format:** Read the inputs from stdin solve the problem and write the answer to stdout...
```python\n# YOUR CODE HERE\n```

**### Answer:** (use the provided format with backticks)

------------------------------------------------------------

**2. Model-Specific System Messages (From `PromptConstants`)**

These messages instruct the model on its role and expected output format.

**DeepSeek-R1 Style:**
```
<|begin of sentence|>A conversation between User and Assistant...
<think> reasoning process here </think> <answer> answer here
</answer>.<|User|>
```

**CodeQwen Style:**
```
<|im_start|>system\nYouare a helpful assistant.
<|im_end|>\n<|im_start|>user
```

------------------------------------------------------------

ating core scientific concepts, or producing initial drafts of sections. All intellectual contributions, analyses, and final content were solely the work of the human authors, who take full responsibility for the paper.

**Question:**
You are given a 0-indexed binary string s of length n on which you can apply two types of operations:
Choose an index i and invert all characters from index 0 to index i (both inclusive), with a cost of i + 1
Choose an index i and invert all characters from index i to index n - 1 (both inclusive), with a cost of n - i
Return the minimum cost to make all characters of the string equal.
Invert a character means if its value is '0' it becomes '1' and vice-versa.

```
class Solution:
    def minimumCost(self, s: str) -> int:
        n = len(s)
        transitions = []
        for i in range(n - 1):
            if s[i] != s[i + 1]:
                transitions.append(i)

        if not transitions:
            return 0

        cost0 = 0  # cost to make current segment 0
        cost1 = float('inf')  # cost to make current segment 1

        for i in transitions:
            prefix_cost = i + 1
            suffix_cost = n - i

            # Calculate new costs based on flipping prefix or suffix
            new_cost0 = min(cost1 + prefix_cost, cost1 + suffix_cost)
            new_cost1 = min(cost0 + prefix_cost, cost0 + suffix_cost)

            cost0, cost1 = new_cost0, new_cost1

        # Check the final segment after the last transition
        last_i = transitions[-1]
        final_length = n - (last_i + 1)
        if final_length > 0:
            final_prefix = last_i + 2
            final_suffix = n - (last_i + 1)
            final_char = s[last_i + 1]
            if final_char == '0':
                # For cost1, need to flip the final segment
                cost1 += min(final_prefix, final_suffix)
            else:
                # For cost0, need to flip the final segment
                cost0 += min(final_prefix, final_suffix)

        result = min(cost0, cost1)

        return result
```

## Code with Properties

```
class Solution:
    def minimumCost(self, s: str) -> int:
        n = len(s)
        transitions = []
        for i in range(n - 1):
            if s[i] != s[i + 1]:
                transitions.append(i)

        if not transitions:
            return 0

        cost0 = 0  # cost to make current segment 0
        cost1 = float('inf')  # cost to make current segment 1

        for i in transitions:
            prefix_cost = i + 1
            suffix_cost = n - i

            # Calculate new costs based on flipping prefix or suffix
            new_cost0 = min(cost1 + prefix_cost, cost1 + suffix_cost)
            new_cost1 = min(cost0 + prefix_cost, cost0 + suffix_cost)

            cost0, cost1 = new_cost0, new_cost1

        ......(13 lines left)......

        result = min(cost0, cost1)

        # Assertion checks to ensure correctness
        assert result >= 0, f"Negative cost detected: {result}"
        if not transitions:
            assert result == 0, f"Uniform string, expected cost 0, got {result}"
        else:
            assert result > 0, "Result should be positive when transitions exist"
        assert not math.isinf(cost0), "cost0 must be finite"
        assert not math.isinf(cost1), "cost1 must be finite"
        return result
```

## Results of Execution

```
{
    "error": "AssertionError('cost0 must be finite')",
    "error_message": "Runtime Error",
    "inputs": "['0011']",
    "expected": "2"
}
```

## Correct Code

```
class Solution:
    def minimumCost(self, s: str) -> int:
        n = len(s)
        total_cost = 0

        # Iterate through the string to find all transition points.
        for i in range(n - 1):
            if s[i] != s[i+1]:
                # A transition exists at index 'i'.
                # Cost to fix it by flipping the prefix s[0...i].
                prefix_flip_cost = i + 1

                # Cost to fix it by flipping the suffix s[i+1...n-1].
                # This is operation 2 starting at index i+1.
                suffix_flip_cost = n - (i + 1)

                # Add the minimum of the two costs to the total.
                total_cost += min(prefix_flip_cost, suffix_flip_cost)

        return total_cost
```

## Model Output

```
<think>
We are given a binary string `s`. We can perform two operations:
...(3292 tokens left)...
</think>
```

Figure 7: **A Case Study on the PGS Workflow.** This figure illustrates the end-to-end refinement process for a challenging problem. (**1**) The process starts with an initial, complex, and incorrect solution. (**2**) This code is then instrumented with property-based assertions. (**3**) Executing the instrumented code with a minimal input triggers a clear `AssertionError`, transforming a latent logic bug into an explicit failure signal. (**4**) This targeted feedback enables the LLM to effectively debug the program, ultimately producing a (**5**) much simpler and correct final solution.

Table 7: **Prompt Templates for Property Generation.** This table illustrates the layered construction of prompts that instruct an LLM to generate Python functions for property validation.

---

**1. Core User Prompt for Property Generation**

This is the base instruction set given to the model, asking it to act as a testing expert.

**### Task Description:**
You are a software testing expert. Your task is to analyze the problem description and generate a Python function that asserts a specific property or invariant a correct solution must satisfy. This property-checking function should take the candidate solution's input and output, returning `True` if the property holds, or `False` if it fails.

**### Inputs Provided for Context:**
- Problem Description: {`question`}
- Example of a Correct Solution's Input/Output: {`example_solution_io`}

------------------------------------------------------------

**2. Property Type-Specific Guidance**

The core prompt is refined with examples based on the type of property required.

**Relational Property Example (e.g., for a sorting problem):**
The property checks the relationship between input and output.

```
# Generated code checks if output is a sorted permutation of input
def check_property(input_list, output_list):
    return sorted(input_list) == output_list
```

**Intrinsic Property Example (e.g., for prime factorization):**
The property checks a characteristic of the output itself.

```
# Generated code checks if the product of factors equals the input
def check_property(n, factors):
    product = 1
    for factor in factors:
        product *= factor
    return product == n
```

------------------------------------------------------------

**3. Model-Specific System Messages & Wrappers**

Finally, the entire prompt is wrapped with model-specific system messages and formatting tags.

**DeepSeek-R1 Style:**
```
<|begin of sentence|>A conversation...  The assistant first
thinks...  <think> reasoning process here </think> <answer>
answer here </answer>.<|User|> [Core Prompt + Property Guidance]
<|Assistant|>
```
**CodeQwen Instruct Style:**
```
<|im_start|>system\nYouare a helpful AI...<|im_end|>
<|im_start|>user\n[Core Prompt + Property Guidance]<|im_end|>
<|im_start|>assistant
```

**Question:**
You are given a 0-indexed integer array receiver of length n and an integer k.
There are n players having a unique id in the range [0, n - 1] who will play a ball passing game, and receiver[i] is the id of the player who receives passes from the player with id i. Players can pass to themselves, i.e. receiver[i] may be equal to i. You must choose one of the n players as the starting player for the game, and the ball will be passed exactly k times starting from the chosen player.
For a chosen starting player having id x, we define a function f(x) that denotes the sum of x and the ids of all players who receive the ball during the k passes, including repetitions.
In other words, f(x) = x + receiver[x] + receiver[receiver[x]] + ... + receiver^(k)[x].

```python
class Solution:
    def getMaxFunctionValue(self, receiver: list[int], k: int) -> int:
        n = len(receiver)
        visited = [False] * n
        p = [0] * n
        sum_p = [0] * n
        m = [0] * n
        sum_c = [0] * n
        cycle_prefix_sums = [[] for _ in range(n)]
        cycle_index = [-1] * n
        for x in range(n):
            if not visited[x]:
                path = []
                current = x
                while True:
                    if visited[current]:
                        if current in path:
                            # Precompute prefix sums for the cycle
                            ......(28 lines left)......
                        else:
                            # Process the tree path leading to a visited node
                            ......(11 lines left)......
                    else:
                        visited[current] = True
                        path.append(current)
                        current = receiver[current]
        max_f = 0
        best_x = -1
        for x in range(n):
            total_steps = k + 1
            if total_steps <= p[x]:
                current_sum = sum_p[x]
            else:
                pre_steps = p[x]
                full_cycles = (total_steps - pre_steps) // m[x]
                remaining_steps = (total_steps - pre_steps) % m[x]
                current_sum = sum_p[x] + full_cycles * sum_c[x]
                if remaining_steps > 0:
                    current_sum +=
cycle_prefix_sums[x][remaining_steps]
            if current_sum > max_f:
                max_f = current_sum
                best_x = x
        return max_f
```

## Code with Properties

```python
class Solution:
    def getMaxFunctionValue(self, receiver: list[int], k: int) -> int:
        ......(57 lines left)......
        max_f = 0
        best_x = -1
        for x in range(n):
            total_steps = k + 1
            if total_steps <= p[x]:
                current_sum = sum_p[x]
            else:
                pre_steps = p[x]
                full_cycles = (total_steps - pre_steps) // m[x]
                remaining_steps = (total_steps - pre_steps) % m[x]
                current_sum = sum_p[x] + full_cycles * sum_c[x]
                if remaining_steps > 0:
                    current_sum +=
cycle_prefix_sums[x][remaining_steps]

            # Property Verification 1: Sum must be non-negative
            assert current_sum >= 0, f"Negative sum for x={x}"

            # Property Verification 2:
            # For small k, computed sum must match simulation
            if k <= 100:
                def simulate_sum(receiver, x, k):
                    total = x
                    current = x
                    for _ in range(k):
                        current = receiver[current]
                        total += current
                    return total

                simulated_sum = simulate_sum(receiver, x, k)
                assert simulated_sum == current_sum,
                    f"Simulated sum {simulated_sum} does not match
computed {current_sum} for x={x}"
            if current_sum > max_f:
                max_f = current_sum
                best_x = x

        return max_f
```

## Correct Code

```python
class Solution:
    def getMaxFunctionValue(self, receiver: List[int], k: int) -> int:
        n = len(receiver); m = 60
        dp = [[0] * n for _ in range(m)]
        dp_sum = [[0] * n for _ in range(m)]
        for i in range(n):
            dp[0][i] = receiver[i]
            dp_sum[0][i] = receiver[i]
        for j in range(1, m):
            for i in range(n):
                mid = dp[j-1][i]
                dp[j][i] = dp[j-1][mid]
                dp_sum[j][i] = dp_sum[j-1][i] + dp_sum[j-1][mid]
        ans = -10**25
        for x in range(n):
            cur = x; s = 0
            for j in range(m):
                if (k >> j) & 1:
                    s += dp_sum[j][cur]
                    cur = dp[j][cur]
            candidate = x + s
            if candidate > ans:
                ans = candidate
        return ans
```

## Results of Execution

```json
{
    "error": "AssertionError('Simulated sum 4 does not
match computed 5 for x=1')",
    "error_message": "Runtime Error",
    "inputs": "[[2, 0, 1], 4]",
    "expected": "6"
}
```

## Model Output

```
<think>
The original code reported an error in
assert simulated_ sum==current sum:
...(4297 tokens left)...
</think>
```

Figure 8: **A Case Study on the PGS Workflow.** This figure illustrates the end-to-end refinement process for a challenging problem. **(1)** The process starts with an initial, complex, and incorrect solution. **(2)** This code is then instrumented with property-based assertions. **(3)** Executing the instrumented code with a minimal input triggers a clear `AssertionError`, transforming a latent logic bug into an explicit failure signal. **(4)** This targeted feedback enables the LLM to effectively debug the program, ultimately producing a **(5)** much simpler and correct final solution.

Table 8: **Prompt Templates for Dynamic Input Script Generation.** This table shows the layered construction of prompts that instruct an LLM to generate a Python script, which in turn produces randomized test inputs.

---

**1. Core User Prompt for Input Script Generation**

This is the base instruction set given to the model, outlining the primary task.

**### Task Description:**
You are an expert Python programmer. Your task is to write a Python script that utilizes randomization (seeded by current time) to generate diverse and valid input strings. The script's standard output must be a single string formatted exactly as required.

**### Inputs Provided for Context:**
- Problem Description: {`question`}
- Original Code Snippet: {`original_code_snippet`}
- Target Platform: {`platform`}
- Example Input String: {`example_input_str`}
- - - - - - - - - - - - - - - - - - - - - - - - - - - - - - - - - - - - - - - - - - - -

**2. Platform-Specific Formatting Guidance**

The core prompt is augmented with specific instructions based on the target platform's input format.

**LeetCode / MBPP / HumanEval Style Example:**
Your Python script should generate a string where each line is a JSON object.

```
Example Target Output String:
[1,2,3]
"some_string"
```

**Codeforces / AtCoder Style Example:**
Your Python script should generate a string with space- or newline-separated values.

```
Example Target Output String:
3
1 2 3
```

- - - - - - - - - - - - - - - - - - - - - - - - - - - - - - - - - - - - - - - - - - - -

**3. Model-Specific System Messages & Wrappers**

Finally, the entire prompt is wrapped with model-specific system messages and formatting tags.

**DeepSeek-R1 Style:**
```
<|begin of sentence|>A conversation...  The assistant first
thinks...  <think> reasoning process here </think> <answer>
answer here </answer>.<|User|> [Core Prompt + Platform Guidance]
<|Assistant|>
```
**CodeQwen Instruct Style:**
```
<|im_start|>system\nYouare a helpful AI...<|im_end|>
<|im_start|>user\n[Core Prompt + Platform Guidance]<|im_end|>
<|im_start|>assistant
```

Table 9: **Prompt Templates for Feedback-Driven Code Repair.** This table outlines the prompt structure for guiding an LLM to debug and fix erroneous code based on specific execution feedback.

---

**1. Core Repair Prompt Structure**

This is the main template that presents the problem, the buggy code, and the specific error context to the model.

**### Question:**
{question}

**### Buggy Code:**
```python\n{buggy_code}\n```

**### Error Context:**
{error_feedback_from_part_2}

**### Your Task:**
First, provide a concise explanation of the error. Then, generate the entire corrected program.
- - - - - - - - - - - - - - - - - - - - - - - - - - - - - - - - - - - - - - - - - - - - - - - - -

**2. Dynamic Error Feedback Generation**

This component translates a structured error object ('metadata') into a human-readable feedback string.

**IF** error_code == -2 (Wrong Answer):
Context: The program previously produced a wrong answer.
Input: {inputs}
Generated Output: {output}
Expected Output: {expected}
**ELSE**:
Context: The program previously encountered a runtime error.
Input: {inputs}
Error Details: {error}
- - - - - - - - - - - - - - - - - - - - - - - - - - - - - - - - - - - - - - - - - - - - - - - - -

**3. Model-Specific System Messages & Wrappers**

Finally, the entire prompt is wrapped with model-specific system messages and formatting tags.

**DeepSeek-R1 Style:**
```
<|begin of sentence|>A conversation...  The assistant first
thinks...  <think> reasoning process here </think> <answer>
answer here </answer>.<|User|> [Core Prompt + Platform Guidance]
<|Assistant|>
```
**CodeQwen Instruct Style:**
```
<|im_start|>system\nYouare a helpful AI...<|im_end|>
<|im_start|>user\n[Core Prompt + Platform Guidance]<|im_end|>
<|im_start|>assistant
```

