# OpenReview forum: "Property-Oriented and Structurally Minimal Feedback for Effective LLM-based Code Refinement"
_ICLR.cc/2026/Conference — ICLR 2026 Conference Withdrawn Submission_

### Official Review · Reviewer_jZ2a · 2025-11-01

**Soundness:** 3
**Presentation:** 3
**Contribution:** 2
**Rating:** 6
**Confidence:** 2

**Summary:**

The paper presents PGS (Property-Generated Solver), a novel framework for LLM-based code refinement that emphasizes feedback quality rather than quantity. By generating property-oriented feedback (semantic correctness conditions) and enforcing structurally minimal counterexamples, PGS improves the interpretability and precision of debugging signals. It establishes a Generator–Tester multi-agent loop and demonstrates state-of-the-art results on multiple benchmarks (HumanEval, MBPP, LiveCodeBench, etc.) without additional training, outperforming self-debugging and repair baselines.

**Strengths:**

- PGS is conceptually original, reframing LLM code debugging around semantic properties and minimal feedback. It is empirically strong, with consistent improvements across diverse datasets and models.

-The approach is training-free, interpretable, and model-agnostic, supported by well-designed ablations and mechanism analyses.

- Clarity and presentation are good, and the work offers both theoretical motivation and practical value for future LLM code refinement systems.

**Weaknesses:**

- Performance depends on the quality of generated properties, and the framework is currently limited to Python settings.

- Efficiency analysis and runtime overhead are insufficiently quantified, and failure cases are underexplored.

- While results are robust, extending evaluations to multi-language or large-scale industrial codebases would better validate scalability and generality.

**Questions:**

- How does PGS handle incorrect or partially valid properties generated by the Tester?

- Have the authors explored dynamic property evolution, where properties are refined across iterations?

- How does runtime cost scale with model size and number of refinement rounds?

---

> ### Author Response · Authors · 2025-11-26
> **Author Rebuttal to Review jZ2a**
>
> ### **W1&Q1: How does PGS handle incorrect or partially valid properties generated by the Tester?**
>
> We address the reliability of generated properties through a rigorous verification mechanism supported by empirical analysis:
> - **Property Validation.** As described in Section 4.2, before any property is used, PGS converts it into an executable assertion and validates it against public test cases. Any property that conflicts with known ground truth is immediately discarded. This acts as a hard filter for incorrect properties.
> - **The Generation-Validation Gap.** To quantify the reliability of this approach, we conduct an additional experiment on LiveCodeBench (using DeepSeek-R1-Distilled-34B). The results show a significant capability gap: on hard problems, the model can only generate **32.4%** correct code, while it (using the same model) can correctly verify **76.5%** solutions. When applying property validation, it further boosts the validation accuracy to **88.2%** (see **Table 3 in revised manuscript**). This indicates generated properties after filtering can effectively provide robust semantic guidance for subsequent refinement processes.
>
> ***
>
> ### **W2: Lack of Efficiency Analysis and Case Studies.**
>
> - **Efficiency Analysis.** We conduct a detailed Cost-Effectiveness Analysis (Details in **Section 5.3.2 in revised manuscript**) to quantify this. PGS offers the optimal trade-off, delivering the highest accuracy with lower computational overhead than its counterparts.
> - **Case Studies.** Case studies can be found in **Appendix C in revised manuscript**.
>
> ***
>
> ### **W1&W3: Extending evaluations to multi-language or large-scale industrial codebases.**
>
> - **Multi-Language Code Generation.** We conduct additional evaluations for C/C++ on HumanEval and LiveCodeBench in **Appendix D.3 in revised manuscript**. The results demonstrate the generalizability of PGS on multiple programming languages.
> - **Large-Scale Industrial Codebases.** We conduct experiments on SWE-bench Verified, a rigorous benchmark for real-world repository-level software engineering. The results are provided in **Table B in revised manuscript**. PGS consistently outperforms all baselines. For example, on Qwen3-Coder-30B, PGS achieves a **50.7%** pass rate on SWE-bench Verified, surpassing the SWE-agent(**46.5%**). This confirms PGS can achieve robust results on large-scale industrial codebases.
>
> ***
>
> ### **Q2: Have the authors explored dynamic property evolution, where properties are refined across iterations?**
>
> PGS generates new properties at each iteration instead of refining previous properties. This is because:
> - A property designed to catch a bug in iteration $i$ often becomes obsolete in iteration $i+1$. After refining the code using the property generated in iteration $i$, the primary error within the code shifts (e.g., from a fundamental logic error to a subtle edge case). Generating new properties ensures the Tester targets the current state of the code rather than refining checks for already-fixed issues.
> - Implementing property evolution requires maintaining and processing a history of property states, which significantly increases token consumption and context usage. Our generation strategy prioritizes computational efficiency, allowing the model to focus its context window on the immediate code logic.
>
> ***
>
> ### **Q3: How does runtime cost scale with model size and number of refinement rounds?**
>
> **A3:** The runtime cost scales linearly with the number of refinement rounds. We conduct a detailed Cost-Effectiveness Analysis (Details in **Section 5.3.2 in the revised manuscript**) to quantify this. PGS offers the optimal trade-off, delivering the highest accuracy with lower computational overhead than its counterparts.

---

> ### Author Response · Authors · 2025-11-28
>
> Dear Reviewer,
> ﻿
>
> Thank you again for your constructive comments on our paper.
> ﻿
>
> With the discussion period ending soon, we wanted to kindly follow up to see if our previous response and the revised manuscript have satisfactorily addressed your concerns.
> ﻿
>
> If our response has clarified your doubts, we would be grateful if you could consider reconsidering your score. Alternatively, if there are still outstanding issues, please let us know so we can address them before the deadline.
> ﻿
>
> Best regards,
> ﻿
>
> Authors

---

### Official Review · Reviewer_DP7V · 2025-11-01

**Soundness:** 2
**Presentation:** 3
**Contribution:** 2
**Rating:** 4
**Confidence:** 4

**Summary:**

This paper proposed a new method called the Property-Generated Solver framework to help LLMs refine code. Instead of just running more tests, PGS focuses on generating higher-quality feedback. It does this by using two key principles: validating the code's core logic and providing minimal, clear counterexamples. This process is powered by a multi-agent system. In experiments across several coding benchmarks, PGS significantly outperformed traditional test-driven development and other modern debugging techniques.

**Strengths:**

- Clear articulation of a novel principle (feedback quality focus).
The paper introduces a conceptually appealing and well-motivated shift in perspective: instead of increasing the number of test cases for LLM code refinement, it emphasizes improving the intrinsic quality of feedback.

- Strong empirical performance on multiple benchmarks. The experiments are extensive and convincing. PGS shows clear improvements on HumanEval, MBPP, LiveCodeBench, and CodeContests, outperforming strong recent methods under identical computational budgets.
- Well-written and structured paper.

**Weaknesses:**

1.**Lack of rigorous theoretical grounding for property-oriented feedback**
The core claim that property-oriented feedback provides more “semantic” guidance is intuitively appealing but lacks formal or theoretical justification.

   (a)The paper does not analyze why LLMs can benefit from property abstraction beyond empirical evidence.

   (b)The concept of “property minimality” (minimizing token count) is purely empirical; there’s no analysis of whether this metric universally correlates with reasoning efficiency or reduced cognitive load.

2.**Dependence on LLM-generated properties introduces circularity**
Although the paper emphasizes avoiding the “self-deception cycle” of LLM-generated tests, the Tester in PGS still relies on an LLM to generate properties and test inputs.

   (a)This means the claimed independence between code generator and feedback generator is only partial—both may share underlying biases, especially when using the same base model family.

   (b)The validation mechanism (checking properties against public tests) may not sufficiently filter out incorrect or incomplete properties.


3.**Evaluation does not isolate contribution of individual components clearly**
The ablation studies mainly compare the overall framework vs. baselines. However, the contribution of specific modules, such as property synthesis, input synthesis, and feedback minimization, is not fully disentangled.

**Questions:**

Q1: Could you elaborate on the theoretical or analytical basis for this assumption?
Specifically, how do property abstraction and feedback minimality improve the model’s reasoning or generalization capability beyond empirical evidence?、

Q2: How do you ensure that the Tester’s feedback is truly independent and reliable?
Can you provide quantitative evidence on the correctness or filtering accuracy of generated properties, or discuss measures (e.g., using distinct models or cross-validation) that prevent shared logical biases between the two agents?

---

> ### Author Response · Authors · 2025-11-26
> **Author Rebuttal to Review DP7V**
>
> ### **W1(a)&Q1: Theoretical justification of property-oriented feedback providing semantic guidance.**
>
> We would like to clarify that our approach is grounded in established software engineering principles, specifically **Metamorphic Testing (MT)** [1-3].
>
> The core challenge in automated code generation/refinement is the test oracle problem. Generating the correct output (the oracle) for a complex test input can be as difficult for an LLM as solving the original problem itself. Instead of verifying a specific, complex input-output pair, Metamorphic Testing was developed precisely to address this challenge by validating **Metamorphic Relations (MRs)**, i.e., fundamental invariants that the program's output must satisfy.
>
> Our property-oriented feedback is a direct application of these MRs. As we demonstrate in **Figure 1** (prime factorization problem):
> *   A simple I/O failure (e.g., input: `99460729`, output: `9973`) is an isolated data point that offers little insight.
> *   In contrast, a property violation (e.g., Failure Feedback 2: `The product of prime numbers is 2 != 4`) reveals the program's core deficiency (the factorization is incomplete).
>
> This is the essence of "semantic guidance": the feedback moves beyond an I/O mismatch to reveal *why* the code is wrong at a fundamental level. This steers the LLM toward a generalizable fix that corrects the underlying logic, rather than an overfitted patch for a single failing test case. Thus, our method's effectiveness is rooted in MT's established success at alleviating the test oracle problem.
>
> [1] A Survey on Metamorphic Testing. Sergio Segura, Gordon Fraser, Ana B. Sanchez, et al. IEEE Transactions on Software Engineering, 2016.
>
> [2] Metamorphic Testing: A Review of Challenges and Opportunities. Tsong Yueh Chen, Fei-Ching Kuo, Huai Liu, et al. ACM Computing Surveys, 2018.
>
> [3] How Effectively Does Metamorphic Testing Alleviate the Oracle Problem? Huai Liu, Fei-Ching Kuo, Dave Towey, et al. IEEE Transactions on Software Engineering, 2014.
>
> ***
>
> ### **W1(b)&Q1: Correlation between "property minimality" and reasoning efficiency or reduced cognitive load.**
>
> We would like to clarify that the "property minimality" is grounded in the principle of **Delta Debugging** [1-3] (We also mention it in the paper, line 220 and 350).
>
> The core theory of Delta Debugging is that a structurally minimal failing input is the most effective signal for bug isolation. It works by systematically removing irrelevant, distracting "noise" from the execution trace to pinpoint the error's root cause.
>
> In our pilot study, we conduct an empirical investigation to determine the best proxy for the "minimality" principle that an LLM would be most sensitive to. The results in **Table 1** showcase that minimizing by "Token Count" consistently and significantly outperformed all other strategies in terms of bug fix rate (Pass@1).
>
> [1] HDD: hierarchical delta debugging. Ghassan Misherghi, Zhendong Su. In Proceedings of the 28th international conference on Software engineering, 2006.
>
> [2] Coarse Hierarchical Delta Debugging. Renáta Hodován, Ákos Kiss, Tibor Gyimóthy, et al. IEEE International Conference on Software Maintenance and Evolution (ICSME), 2017.
>
> [3] Extending Hierarchical Delta Debugging with Hoisting. Dániel Vince, Renáta Hodován, Daniella Bársony, et al. IEEE/ACM International Conference on Automation of Software Test (AST), 2021.

---

> ### Author Response · Authors · 2025-11-26
> **Author Rebuttal to Review DP7V continued**
>
> ### **W2(a): How our PGS (rely on LLM-generated properties) breaks the circularity?**
>
> Our framework is designed to break this circularity by leveraging a fundamental principle: the **Asymmetry of Verification** [1]. It is widely recognized that verifying a solution's correctness is often a significantly easier and computationally simpler task than generating that solution from scratch.
>
> The "cycle of self-deception" our paper addresses occurs when the test generator must solve the exact same complex problem as the code generator (i.e., generating a correct I/O oracle). Our framework breaks this cycle because the Tester's task is fundamentally simpler than the Generator's task.
> *   **Example:** The Generator must produce a complex factorization algorithm. The Tester, however, only needs to generate a simple, independent property check, such as `assert math.prod(factors) == N`.
> *   **Breaking the Bias:** The risk of an LLM failing to generate this simple assertion (a verification task) is far lower than its risk of failing to generate the complex algorithm (a generation task). This vast difference in task complexity breaks the "shared bias" circularity, as the Tester is operating at a much higher level of reliability.
>
> To provide rigorous empirical evidence for this asymmetry, we conduct a new experiment on a 100-problem subset of LiveCodeBench (32 easy, 34 medium, 34 hard) in **Table 3 and Appendix D.1 in the revised manuscript**. We compare the LLM's success rate at **generation** (solving the problem from scratch) versus its success rate at **verification** (writing a verifier that correctly identifies correct and incorrect solutions).
>
> Our results empirically confirm this asymmetry of verification. The accuracy of verification (**wo/ filtering 87.0%, w/ filtering 93.0%**) is significantly higher than that of code generation (**63.0%**). Especially when dealing with hard problems, the verification accuracy (**wo/ filtering 76.5%, w/ filtering 88.2%**) is more than twice the generation accuracy (**32.4%**). This confirms the Tester's task is easier and more reliable.
>
> [1] Asymmetry of Verification and Verifier's Law. Jason Wei. https://www.jasonwei.net/blog/asymmetry-of-verification-and-verifiers-law
>
> ***
>
> ### **W2(b): Robustness of Property Validation Mechanism (checking properties against public tests).**
>
> Our methodology mitigates this by generating multiple candidate properties and using the public tests as a robust filter. As shown in **Table 3 in the revised manuscript**, our filtering mechanism is highly effective. It boosts the verification accuracy of hard problems from **76.5% to 88.2%**, widening the reliability gap over the **32.4%** generation accuracy even further.
>
> ***
>
> ### **W3: Ablation on Contribution of Each Component.**
>
> We conduct an additional ablation study on LiveCodeBench (using DeepSeek-R1-Distilled-32B) to investigate the contribution of each component in PGS. The results are shown in **Table 4 in the revised manuscript**.
>
> *   **Structural Minimization Feedback (+2.8%).** Simply switching the feedback mechanism from reporting a random failing input to reporting the minimal failing input (based on token count) improves the pass rate from 64.4% to 67.2%. This validates our hypothesis that even in the absence of properties, reducing the cognitive load of feedback signals effectively enhances the model's debugging capability.
> *   **Property Generation (+1.3%) & Validation (+3.1%).** Transitioning from I/O-based feedback to Property-Oriented feedback yields a modest gain. This suggests that the quality of properties generated by LLMs may vary significantly, potentially containing hallucinations or logical flaws. Applying the property validation mechanism provides a significant boost, indicating that using high-quality properties after filtering to generate feedback can effectively provide robust semantic guidance for subsequent refinement processes.
> *   **Iterative Refinement (+4.9%).** Finally, allowing the LLM to engage in a multi-round refinement loop further improves performance to 76.5%. This confirms that complex bugs often require stepwise correction rather than a single-shot fix.
>
> ***
>
> ### **Q2: How do you ensure that the Tester's feedback is truly independent and reliable? Can you provide quantitative evidence on the correctness or filtering accuracy of generated properties, or discuss measures (e.g., using distinct models or cross-validation) that prevent shared logical biases between the two agents?**
>
> *   **Tester's Independency and Reliability:** Please refer to **W2(a)**.
> *   **Quantitative Evidence on Tester's Reliability:** Please refer to **W2(b)**.

---

> ### Author Response · Authors · 2025-11-28
>
> Dear Reviewer,
> ﻿
>
> Thank you again for your constructive comments on our paper.
> ﻿
>
> With the discussion period ending soon, we wanted to kindly follow up to see if our previous response and the revised manuscript have satisfactorily addressed your concerns.
> ﻿
>
> If our response has clarified your doubts, we would be grateful if you could consider reconsidering your score. Alternatively, if there are still outstanding issues, please let us know so we can address them before the deadline.
> ﻿
>
> Best regards,
> ﻿
>
> Authors

---

### Official Review · Reviewer_bs6j · 2025-11-01

**Soundness:** 3
**Presentation:** 3
**Contribution:** 2
**Rating:** 4
**Confidence:** 4

**Summary:**

This paper targets the persistent gap in functional correctness of LLM-generated code. It argues that the efficacy of TDD-style repair is constrained by feedback quality, which is undermined by the scarcity of high-quality tests and the noise of automatically generated suites. The central thesis is quality over quantity in feedback, instantiated via two principles: property-oriented and structurally minimal. Property-oriented feedback verifies general semantic properties that correct programs must satisfy (e.g., sorted output is non-decreasing), providing oracle-agnostic guidance beyond input–output mismatches and mitigating the test-oracle problem. Structural minimality demands the simplest counterexample that triggers a property violation, lowering cognitive load and isolating root causes. To realize these principles, the authors propose Property-Generated Solver (PGS), a multi-agent framework with a Generator (for code synthesis and repair) and a Tester (for high-quality feedback). The Tester extracts properties from problem statements, generates diverse inputs to probe them, detects violations, and returns the structurally simplest counterexample as a clear corrective signal. Across HumanEval, MBPP, LiveCodeBench, and CodeContests, PGS consistently outperforms TDD and self-debugging baselines on multiple LLMs, raising bug-repair rates by 1.4×–1.6× and setting a new state of the art.

**Strengths:**

1. **Novel and Insightful Core Idea**: The paper's primary contribution is its novel and insightful perspective that shifts the research paradigm from test quantity to feedback quality. This addresses a critical yet largely overlooked dimension in existing work on LLM-based code refinement, providing a valuable new direction for the field.

2. **Rigorous and Well-Grounded Methodology**: The methodology is rigorous and logically sound. A key highlight is the well-designed pilot study that empirically validates the paper's foundational principles before introducing the full framework. This study convincingly demonstrates the superiority of property-oriented feedback over simple I/O mismatches and identifies input token count as the most effective proxy for creating structurally minimal, actionable feedback.

3. **Comprehensive and Solid Empirical Evaluation**: The paper is supported by a comprehensive and convincing empirical evaluation. The experiments are conducted across a diverse suite of widely-recognized benchmarks, spanning a wide spectrum of difficulty from function-level synthesis to competition-level programming. The validation on multiple state-of-the-art language models further demonstrates the robustness and generalizability of the proposed framework, showcasing its model-agnostic benefits.

**Weaknesses:**

1. **Potentially Outdated Foundation Models**: The generalizability of the findings is potentially limited by the choice of foundation models (DeepSeek-Coder-V2, Qwen2.5-Coder, DeepSeek-R1-Distilled-32B). These models, while capable, are no longer at the state-of-the-art. The performance gap between the strongest model used and more recent, powerful models (e.g., Qwen3-Coder-30B, which scores over 50 on SWE-bench) is significant. This raises questions about whether the observed gains would persist on leading proprietary models (e.g., GPT-5, Claude 4 series), which may exhibit different failure modes and require different feedback strategies.

2. **Questionable Cost-Effectiveness on Simple Tasks**: The application of a sophisticated, multi-agent framework like PGS to single-function benchmarks (e.g., HumanEval, MBPP) raises questions of cost-effectiveness. Although performance gains are reported, these benchmarks are often solvable by simpler methods. The results are therefore less significant in justifying the framework's complexity, as its true advantage lies in tackling more intricate problems where such sophistication is necessary.

3. **Unaddressed Dependence on Tester Agent Capability**: The framework's success appears to be heavily dependent on the capability of the Tester agent, yet its robustness is insufficiently discussed. The paper does not adequately address scenarios where the Tester might generate incorrect or trivial properties, which could misguide the refinement process and degrade performance. An analysis of the framework's sensitivity to the Tester's capability (e.g., by using a weaker model) would strengthen the claims.

4. **Lack of Cost-Benefit Analysis**: The paper lacks a thorough cost-benefit analysis. The multi-step iterative process of PGS likely incurs substantial computational overhead (in terms of token usage and latency) compared to simpler refinement methods. Without a direct comparison under a fixed computational budget, it is difficult to assess the practical trade-offs and overall efficiency of the proposed approach.

5. **Discrepancy in Reported Baseline Score**: A minor discrepancy was noted in the baseline performance for DeepSeek-R1-Distilled-32B on LiveCodeBench. The paper reports a pass@1 of 64.4%, which appears to differ from the 57.2% cited in the model's official technical report. Clarification on this point would be beneficial for ensuring the reproducibility of the results.

6. **Minor Presentation Flaw in Appendix**: There is a presentation error in the appendix where Figure 6 and Figure 7 are identical. This appears to be a copy-paste error and should be corrected to reflect the distinct case studies described in the text.

**Questions:**

1. Generalizability to State-of-the-Art Models: The paper demonstrates strong results on capable open-source models. However, to what extent do the benefits of property-oriented and structurally minimal feedback generalize to leading proprietary models (e.g., the GPT-5 ，Claude 4 series and Qwen3-Coder)? A supplementary analysis on these models would significantly bolster the paper's claims of generalizability.

2. Sensitivity to the Tester Agent's Capability: The framework's performance seems contingent on a highly capable Tester agent. How sensitive is the overall performance of PGS to the capability of this agent? Specifically, what is the impact on the fix rate if a weaker model (e.g., the same model as the Generator) is used for property and input synthesis?

3. Efficiency Under a Fixed Computational Budget: Could you provide a comparison of PGS against other iterative baselines under a fixed computational budget (e.g., total token consumption)? This would provide valuable insights into the practical cost-effectiveness and efficiency of the proposed framework.

4. Clarification on Baseline Performance: There appears to be a discrepancy in the reported baseline score for DeepSeek-R1-Distilled-32B on LiveCodeBench compared to its official technical report. Could you please clarify the reason for this difference (e.g., variations in the evaluation setup, prompting strategy, or model version)?

---

> ### Author Response · Authors · 2025-11-26
> **Author Rebuttal to Review bs6j**
>
> ### **W1&Q1: Experimental Results Using the SOTA Coder Model.**
>
> We conduct experiments using **Qwen3-Coder-30B**, **DeepSeek-V3.1**, and **Claude-4-Sonnet**. These results are presented in **Table 2 of the revised manuscript**. PGS consistently improves performance across all these SOTA models on all benchmarks. The principles of property-oriented feedback are model-agnostic and continue to provide significant gains even as base model capabilities increase.
>
> ***
>
> ### **W2: Questionable Cost-Effectiveness on Simple Tasks: The application of a sophisticated, multi-agent framework like PGS to single-function benchmarks (e.g., HumanEval, MBPP) raises questions of cost-effectiveness. The results are therefore less significant in justifying the framework's complexity, as its true advantage lies in tackling more intricate problems where such sophistication is necessary.**
>
> We conduct a cost-effectiveness analysis on LiveCodeBench, a challenging competitive programming benchmark, often requiring several rounds of code refinement. To accurately compare the amount of compute with other iterative methods, we compare the number of tokens versus final performance trade-off in **Section 5.3.2 in the revised manuscript**. Specifically, on LiveCodeBench:
>
> - **Comparison with Simple Baselines (e.g., Self-Edit).** Although PGS consumes more tokens (1.6x) per iteration due to the property generation step, this cost is quickly offset by its superior performance gains. It is clearly illustrated in **Fig.6 (in revised manuscript)** that PGS after only 2 iterations (\~74.9% Pass@1 at \~13k tokens) already surpasses the final performance of the Self-Edit after 5 full iterations (\~73.5% Pass@1 at \~17k tokens) with fewer token costs.
>
> - **Comparison with Complex Step-by-Step Debugging Baselines (e.g., Self-Debugger).** PGS is significantly more efficient than these methods, both on Pass@1 and token costs. Self-Debugger (\~72.5% Pass@1 at \~44.2k tokens) requires nearly 4 times more tokens to achieve the same performance as PGS (\~72.5% Pass@1 at \~9.7k tokens).
>
> In summary, PGS outperforms other iterative methods in both effectiveness and efficiency (token usage), suggesting the superiority of PGS on challenging code generation/repairing tasks like LiveCodeBench.
>
> ***
>
> ### **W3: Unaddressed Dependence on Tester Agent Capability: The framework's success appears to be heavily dependent on the capability of the Tester agent, yet its robustness is insufficiently discussed. The paper does not adequately address scenarios where the Tester might generate incorrect or trivial properties, which could misguide the refinement process and degrade performance.**
>
> **Robustness of Tester in PGS.**
> The Tester in PGS is robust because PGS adheres to the principle of **Asymmetry of Verification** [1], i.e., verifying a solution's correctness is often a significantly easier and computationally simpler task than generating that solution from scratch.
> *   **Example:** The Generator must produce a complex factorization algorithm. The Tester, however, only needs to generate a simple, independent property check, such as `assert math.prod(factors) == N`.
> *   **Verification vs. Generation:** As demonstrated in **Table 3 in the revised manuscript**, the LLM's success rate at verification (writing a verifier that correctly identifies correct and incorrect solutions) is significantly higher than generation (solving the problem from scratch). This confirms the Tester's task is easier and more reliable.
>
> **PGS's Handling of Incorrect/Trivial Properties.**
> *   **Property Validation:** As described in **Section 4.2**, before any property is used, PGS converts it into an executable assertion and validates it against public test cases. Any property that conflicts with known ground truth is immediately discarded. This acts as a hard filter for incorrect properties.
> *   **The Generation-Validation Gap:** To quantify the reliability of this approach, we conduct an additional experiment on LiveCodeBench. The results show a significant capability gap: on hard problems, the model can only generate **32.4%** correct code, while it (using the same model) can correctly verify **76.5%** solutions. When applying property validation, it further boosts the validation accuracy to **88.2%** (see **Table 3 in revised manuscript**). This indicates generated properties after filtering can effectively provide robust semantic guidance for subsequent refinement processes.
>
> [1] Asymmetry of Verification and Verifier's Law. Jason Wei. https://www.jasonwei.net/blog/asymmetry-of-verification-and-verifiers-law

---

> ### Author Response · Authors · 2025-11-26
> **Author Rebuttal to Review bs6j Continued**
>
> ### **W4&Q3: Lack of Cost-Benefit Analysis: The multi-step iterative process of PGS incurs substantial computational overhead. A comparison under a fixed computational budget is needed to assess the practical trade-offs.**
>
> - **Cost-Benefit Analysis:** Please refer to **W2**. The cost-effectiveness analysis on LiveCodeBench suggests PGS's robustness in both effectiveness and efficiency. We also conduct the cost-effectiveness analysis on HumanEval and reach the same conclusion (see **Figure 6 in revised manuscript**).
> - **Comparison under a Fixed Computational Budget:** Besides, as shown in **Figure 6 in the revised manuscript**, at each point on the horizontal axis representing cumulative token consumption, PGS consistently outperforms other iterative baselines given the same computational budget.
>
> ***
>
> ### **W5&Q4: The Discrepancy in Reported Results of DeepSeek-R1 on LiveCodeBench.**
>
> **Discrepancy Stems from Different Data Version.**
> For instance, the Qwen2.5-Coder [1] evaluated the LiveCodeBench subset from **May 2023 to September 2024**, while DeepSeek-R1 [2] evaluated the LiveCodeBench subset **since August 2024**. We utilize the standard LiveCodeBench v5 full set (880 problems) across all models.
>
> **Re-Evaluation on the Same Subset.**
> To address your concerns, we re-evaluated the same **August-2024 subset** used in the DeepSeek-R1 technical report, with results provided in **Table A** below. The baseline score (**57.8%**) matches the official report, and PGS further improves it to **69.8%**.
>
> **Table A. Re-Evaluation on LiveCodeBench**
> *(subset since Aug-2024, aligned with DeepSeek-R1 technical report. All results are reported using mean ± standard deviation from five independent runs.)*
>
> | Method | DeepSeek-Coder-V2 | Qwen2.5-Coder | DeepSeek-R1-Distilled-32B |
> | :--- | :---: | :---: | :---: |
> | **Baseline** | 22.5 ± 1.5 | 28.2 ± 1.5 | 57.8 ± 1.6 |
> | **CoT** | 22.8 ± 1.4 | 28.5 ± 1.4 | 57.8 ± 1.5 |
> | **Code-T** | 24.5 ± 2.2 | 30.5 ± 2.3 | 63.5 ± 1.9 |
> | **Self-Edit** | 25.8 ± 2.1 | 31.5 ± 1.8 | 66.2 ± 1.8 |
> | **Self-Debug** | 26.9 ± 1.9 | 34.2 ± 1.9 | 65.5 ± 2.0 |
> | **PGS (ours)** | **30.5 ± 1.7** | **37.8 ± 1.6** | **69.8 ± 1.9** |
>
> [1] Yuxiang Huang, et al. "Qwen2.5-Coder Technical Report." arXiv preprint arXiv:2409.12186 (2024).
>
> [2] DeepSeek-AI, et al. "DeepSeek-R1: Incentivizing Reasoning Capability in LLMs via Reinforcement Learning." arXiv preprint arXiv:2501.12948 (2025).
>
> ***
>
> ### **W6: There is a presentation error in the appendix (Figure 6 and 7 are identical).**
>
> **A5:** Thanks for your advice. We have corrected it in the revised manuscript.
>
> ***
>
> ### **Q2: Sensitivity to the Tester Agent's Capability. The framework's performance seems contingent on a highly capable Tester agent. How sensitive is the overall performance of PGS to the capability of this agent? Specifically, what is the impact on the fix rate if a weaker model (e.g., the same model as the Generator) is used for property and input synthesis?**
>
> To clarify, in all experiments reported in the paper, the Generator and the Tester agents are implemented using the same LLM (e.g., DeepSeek-R1-Distilled-32B is used for both roles).
> Besides, as discussed in **W3**, the Tester's task is fundamentally simpler than the Generator's task. Considering the huge generation-to-verification gap (see **Table 3 in revised manuscript**), the framework's performance is not highly contingent on the Tester's capability.

---

> ### Author Response · Authors · 2025-11-28
>
> Dear Reviewer,
> ﻿
>
> Thank you again for your constructive comments on our paper.
> ﻿
>
> With the discussion period ending soon, we wanted to kindly follow up to see if our previous response and the revised manuscript have satisfactorily addressed your concerns.
> ﻿
>
> If our response has clarified your doubts, we would be grateful if you could consider reconsidering your score. Alternatively, if there are still outstanding issues, please let us know so we can address them before the deadline.
> ﻿
>
> Best regards,
> ﻿
>
> Authors

---

### Official Review · Reviewer_N9kz · 2025-11-10

**Soundness:** 3
**Presentation:** 3
**Contribution:** 2
**Rating:** 4
**Confidence:** 4

**Summary:**

This paper introduces Property-Generated Solver (PGS) which provides minimal high-level debug feedback for LLM code refinement. The key insights are: 1) high-level property-based feedback is more effective than raw feedback containing the exact buggy outputs; 2) minimal feedback is more effective to reduce the cognitive load for an LLM. PGS follows these two principles to generate key properties a correct program should satisfy then generate test inputs for those properties and select the minimal ones. Across 4 code generation benchmarks with 3 LLMs, PGS show improved pass rates compared to other iterative refinement baselines.

**Strengths:**

1. This paper is well-written is easy to follow. The two principles: property-oriented and structurally minimal, are well presented with experiments to back up their advantages.

2. The empirical results span multiple benchmark and model combinations, providing evidence that PGS is a general method for code refinement.

**Weaknesses:**

1. While the evaluation metric is named "pass@1", there are some subtleties worth discussing. Iterative refinement methods generate multiple programs so calling it pass@1 is misleading. When comparing multiple methods, a proper measure to control for the amount of compute is necessary, for example, the number of tokens. For example, PGS generates 5 candidate properties and 64 new test inputs in each iteration step. Properly counting the number of tokens would provide a more complete compute vs performance tradeoffs.

2. The benchmarks are small scale code generation. SWE-bench is used only to demonstrate the advantage of structurally minimal feedback. Including experiment results on SWE-bench for PGS would make the paper stronger. The three models used in the experiments include two that are released last year and I would not call them "state-of-the-art" (line 161). Incorporating more recent open-weight models such as Deepseek-v3 and Qwen3 family would strengthen the relevance of the work.

**Questions:**

1. Similar to the discussion above about "pass@1", Figure 2 needs a more nuanced discussion as the two feedback methods include 2 generated programs.

2. Please add error bars to the main results.

---

> ### Author Response · Authors · 2025-11-26
> **Author Rebuttal to Review N9kz**
>
> ### **W1: Pass@1 for iterative refinement methods may be misleading. To measure the actual amount of compute, properly counting the number of tokens would provide a more complete compute vs performance tradeoffs.**
>
> We conduct a cost-effectiveness analysis with iterative methods based on "Pass@1" vs. "Token Cost". As shown in **Section 5.3.2 in the revised manuscript**, we measure the accumulated token consumption versus Pass@1 across 5 iterations.
>
> - **Comparison with Simple Baselines (e.g., Self-Edit).** Although PGS consumes more tokens (1.6x) per iteration due to property generation step, this cost is quickly offset by its superior and performance gains. It is clearly illustrated in Fig.6 (in revised manuscript) that PGS after only 2 iterations (\~74.9% Pass@1 at \~13k tokens on LiveCodeBench, \~85.9% Pass@1 at \~6k tokens on HumanEval) already surpasses the final performance of the Self-Edit after 5 full iterations (\~73.5% Pass@1 at \~17k tokens on LiveCodeBench, \~81.9% Pass@1 at \~6.1k tokens on HumanEval) with fewer token costs.
>
> - **Comparison with Complex Step-by-Step Debugging Baselines (e.g., Self-Debugger).** PGS is significantly more efficient than these methods, both on Pass@1 and token costs. Self-Debugger (\~72.5% Pass@1 at ~44.2k tokens on LiveCodebench, \~83.7 Pass@1 at \~21.5k tokens on Human Eval) requires nearly 4 times more tokens to achieve the same performance as PGS (\~72.5% Pass@1 at \~9.7k tokens on LiveCodebench, \~82.9 Pass@1 at ~4k tokens on Human Eval).
>
> In summary, PGS offers the optimal trade-off, delivering the highest accuracy with lower computational overhead than its counterparts.
>
> ***
>
> ### **W2(a): Experiments on the SWE-bench.**
>
> We extend our evaluation to SWE-bench Verified. Detailed results and configurations can be found in **Table 2 and Appendix B in the revised manuscript**.
>
> - **Experiment Setup.**
>   - **Baselines:** A typical evaluation approach involves using an LLM as the backbone of SWE-agent [1] for code repair. SWE-agent operates on a ReAct loop, guiding the LLM through a structured workflow: codebase analysis, issue reproduction, code editing, and fix verification.
>   - **PGS Implementation:** To evaluate PGS on the SWE-bench, we introduce a key modification to the second step (issue reproduction) of the SWE-agent. Specifically, PGS generates properties based on the issue and integrates them with the relevant code to detect property violations. This helps identify the root cause of the error and provides actionable feedback, offering semantic guidance for subsequent code repair.
>   - **Scale & Metrics:** To thoroughly evaluate generalizability, we test PGS using six base models of varying scales and capabilities. All reported results are the mean of five independent runs, accompanied by their standard deviations.
>
> - **Experiment Results.**
>   We provide a result overview in **Table A** below. PGS consistently outperforms all baselines. For example, on Qwen3-Coder-30B, PGS achieves a **50.7%** pass rate on SWE-bench Verified, surpassing the SWE-agent (**46.5%**). This confirms PGS can achieve robust results on complex repository-level tasks.
>
> **Table A. Pass Rate Comparison on SWE-bench Verified**
>
> | Method | DeepSeek-Coder-V2 | Qwen2.5-Coder | DeepSeek-R1-Distilled-32B | Qwen3-Coder-30B | DeepSeek-V3.1 | Claude-sonnet-4 |
> | :--- | :---: | :---: | :---: | :---: | :---: | :---: |
> | **SWE-Agent** | 9.8 ± 1.4 | 10.3 ± 0.8 | 34.4 ± 2.0 | 46.5 ± 1.7 | 54.2 ± 1.8 | 65.5 ± 1.6 |
> | **PGS (ours)** | **11.9 ± 1.0** | **12.8 ± 0.5** | **37.3 ± 2.3** | **50.7 ± 1.9** | **58.4 ± 2.5** | **70.2 ± 1.5** |
>
> [1] John Yang, et al. "SWE-agent: Agent-Computer Interfaces Enable Automated Software Engineering." NeurIPS 2024.
>
> ***
>
> ### **W2(b): Experimental Results Using the SOTA Coder Model.**
>
> We conduct experiments using Qwen3-Coder, DeepSeek-V3.1, and Claude-4-Sonnet. These results are presented in **Table 2 of the revised manuscript**. PGS consistently improves performance across all these SOTA models on all benchmarks. The principles of property-oriented feedback are **model-agnostic** and continue to provide significant gains even as base model capabilities increase.

---

> ### Author Response · Authors · 2025-11-26
> **Author Rebuttal to Review N9kz Continued**
>
> ### **Q1: Figure 2 needs a more nuanced discussion as the two feedback methods include 2 generated programs.**
>
> We would like to clarify that Fig.2 is designed to reveal the performance difference between two feedback methods ("One-shot I/O Feedback" vs. "One-shot Property Feedback") in one refinement step.
> - Both methods start from the same initial code (the "Direct Prompting" in Fig.2).
> - The computational cost is similar for both methods (each generates only one additional refined program).
> - We do not sample multiple candidates to select the best.
> Therefore, in Fig.2, comparing Pass@1 is sufficient and fair.
>
> ***
>
> ### **Q2: Error Bars of Main Results.**
>
> We update all results in the main results (**Table 2 in the revised manuscript**) to include the mean and standard deviation obtained from five independent runs. The results demonstrate that our method is relatively stable.

---

> ### Author Response · Authors · 2025-11-28
>
> Dear Reviewer,
> ﻿
>
> Thank you again for your constructive comments on our paper.
> ﻿
>
> With the discussion period ending soon, we wanted to kindly follow up to see if our previous response and the revised manuscript have satisfactorily addressed your concerns.
> ﻿
>
> If our response has clarified your doubts, we would be grateful if you could consider reconsidering your score. Alternatively, if there are still outstanding issues, please let us know so we can address them before the deadline.
> ﻿
>
> Best regards,
> ﻿
>
> Authors

---

### Author Response · Authors · 2025-12-03
**Summary of Revisions: Consensus on Novelty, SOTA Upgrades, and Mechanism Validation**

We sincerely thank the Area Chair and all reviewers (N9kz, bs6j, DP7V, jZ2a) for their constructive criticism. We are encouraged that **all reviewers recognize the novelty and value of our core contribution:** shifting the paradigm from "test quantity" to "feedback quality" (via Property-Oriented and Structurally Minimal feedback).

Given the consensus on novelty, our rebuttal focused on addressing concerns regarding experimental comprehensiveness, cost-effectiveness, and theoretical grounding. We have performed extensive additional experiments to resolve these issues. Below is a summary of the key concerns raised and our corresponding actions:

**1. Concern: Applicability of State-of-the-Art (SOTA) Coder Models**

- **Question (N9kz W2, bs6j W1&Q1)**: The initial evaluation relied on older models. Does PGS still provide gains on the latest, stronger models?

- **Our Response**: We extend our evaluation to include **Qwen3-Coder**, **DeepSeek-V3.1**, and **Claude-4-Sonnet** in Table 2 in revised manuscript. PGS consistently establishes new SOTA performance across all these models on all benchmarks, proving our method is model-agnostic and effective even for top-tier LLMs.

**2. Concern: Generalization to Complex & Real-World Tasks**

- **Question (N9kz W2, jZ2a W1&W3)**: Is PGS effective on repository-level tasks or languages beyond Python?

- **Our Response**:

  - **Repo-Level**: We evaluate PGS on **SWE-bench Verified** in Table 2 in revised manuscript. PGS (w/ Qwen3-Coder-30B) achieves a **50.7% pass rate**, significantly outperforming the SWE-agent baseline (46.5%).

  - **Multi-Lingual**: We conduct experiments on **C++ versions** of HumanEval and LiveCodeBench in Appendix D.3 in revised manuscript. PGS improved pass rates by large margins (e.g., +12.3% on LCB-C++), demonstrating strong cross-language generalization.

**3. Concern: Cost-Effectiveness and Efficiency**

- **Question (N9kz W1, bs6j W2&W4&Q3, jZ2a W2&Q3)**: Does the iterative nature of PGS incur prohibitive token costs? Is "Pass@1" a fair metric?

- **Our Response**: We conduct a rigorous **Token Cost vs. Pass Rate** analysis in Section 5.3.2 in revised manuscript. The results demonstrate the effectiveness and efficiency of PGS. For example, PGS surpasses the final performance of "Self-Edit" (5 iterations) in just **2 iterations with lower total token consumption**. Moreover, PGS is proven to be more cost-effective than complex debugging baselines (e.g., Self-Debugger).

**4. Concern: Theoretical Grounding & Tester Reliability**
- **Question (bs6j W3&Q2, DP7V W1&W2&Q1&Q2)**: Does the Tester agent share the same biases as the Generator (circularity)? What if the generated properties are wrong?

- **Our Response**: We introduce the **"Asymmetry of Verification" analysis** (Appendix D.1) and **"Metamorphic Testing" Theory** (Response to Reviewer DP7V Q1&W1). Empirical data shows the model's verification accuracy (writing correct assertions) is significantly higher than its generation accuracy (solving the problem). For hard problems, verification accuracy (88.2%) is nearly **2.7x higher** than generation accuracy (32.4%). This gap ensures the Tester provides reliable, independent guidance, breaking the "cycle of self-deception."

**5. Concern: Component Contribution & Baselines**
- **Question (DP7V W3, N9kz W5&Q4)**: Are the baselines fair? What is the contribution of each module?
- **Our Response**: We conduct a detailed ablation study in Table 4 in revised manuscript, quantifying the gains from Structural Minimization (+2.8%), Property Generation (+1.3%), and Validation Filtering (+3.1%). We clarified the baseline discrepancy on LiveCodeBench (due to dataset versioning) and re-ran baselines to ensure a strictly fair comparison.


**Conclusion**: We have significantly strengthened the paper by **adding 3 SOTA models, 1 repo-level benchmark, multi-lingual support, and deep mechanism analyses**. We believe we have addressed all weaknesses raised. We respectfully ask the AC and reviewers to consider these comprehensive improvements and the paper's undisputed novelty in their final assessment.

---

### Note · Authors · 2026-01-29

**Comment:**

Dear Area Chairs and Reviewers,
﻿

We thank the AC and all reviewers for their insightful feedback and for recognizing the novelty of our Property-Oriented and Structurally Minimal feedback paradigm. After considering the reviews and the meta-review, we have decided to withdraw the paper to further refine it for another venue.
﻿

While we appreciate the consensus on our novelty, we believe the current manuscript requires a more rigorous treatment of the following points:

- Robustness of Verification: We will strengthen our evidence for the "Asymmetry of Verification" to more effectively address concerns regarding shared biases between the generator and tester.
- Evaluation Scale: We plan to expand our experiments to the full SWE-bench and a wider range of benchmarks to demonstrate broader generalization.
- Efficiency Analysis: We will more prominently integrate the cost-effectiveness trade-offs into our core discussion.

We believe these revisions will significantly strengthen the work. Thank you for your constructive guidance.
﻿

Sincerely,

The Authors

**Withdrawal Confirmation:**

I have read and agree with the venue's withdrawal policy on behalf of myself and my co-authors.

---

### Meta-Review · Area_Chair_oqiy · 2026-01-04

**Summary:**

This work investigates how to provide better feedback to LLMs. It proposes a novel paradigm called "Property-Guided Solver" (PGS), which assists LLMs in obtaining external feedback by extracting high-level properties from verification programs and then offering the simplest failure feedback.

Reviewers find the core idea of this paper novel and insightful, and the empirical evaluation comprehensive and reliable.

The reviewers raised the following concerns:

1. The scope and details of the experiments need supplementation. Testing was only conducted on a subset of SWE-Bench, and there is a lack of ablation studies analyzing the independent contributions of each component.

2. The cost-effectiveness analysis is insufficient, requiring additional experiments to evaluate the extra costs associated with the method.

3. Reviewers expressed concerns about the robustness of the proposed method. Since both the generator and the tester are based on LLMs, there may be shared underlying biases, weakening the claim of feedback independence. Additionally, there is a risk of high dependency on the tester’s ability to generate high-quality properties and test inputs.

**Reviewer Concerns:**

In response, the author supplemented experiments with more LLMs and benchmarks and analyzed runtime efficiency, effectively addressing the reviewers' concerns. However, the concern regarding robustness remains a significant risk.

**Reviewer Scores:**

After thorough discussion, reviewer N9kz would raise their score to 6, while the other reviewers are inclined to keep their scores unchanged.

---

### Decision · Program_Chairs · 2026-01-26

Reject